# Impact of dense water flow over a sloping bottom on open-sea circulation: Laboratory experiments and the Ionian Sea (Mediterranean) example

Miroslav Gačić[1], Laura Ursella[1], Vedrana Kovačević[*1], Milena Menna[1], Vlado Malačič[2], Manuel Bensi[1], Maria-Eletta Negretti[3], Vanessa Cardin[1], Mirko Orlić[4], Joël Sommeria[3], Ricardo Viana Barreto[5], Samuel Viboud[3], Thomas Valran[3], Boris Petelin[2], Giuseppe Siena[1], Angelo Rubino[5]

[1] National Institute of Oceanography and Applied Geophysics - OGS, Borgo Grotta Gigante 42/C, Sgonico (TS), 34010, Italy

[2] National Institute of Biology, Marine Biology Station, Fornače 41, Piran, 6330, Slovenia

[3] LEGI, CNRS UMR5519, University of Grenoble Alpes, Grenoble, 1209-1211 rue de la piscine, Domaine Universitaire, Saint Martin d'Hères, 38400, France

[4] Andrija Mohorovičić Geophysical Institute, Faculty of Science, University of Zagreb, Horvatovac 95, Zagreb, 10000, Croatia

[5] University Ca' Foscari of Venice, Dept. of Environmental Sciences, Informatics and Statistics, Via Torino 155, Mestre, 30172, Italy

*Correspondence to*: Vedrana Kovačević (vkovacevic@inogs.it)

**Abstract.** The North Ionian Gyre (NIG) displays prominent inversions on decadal scales. We investigate the role of internal forcing induced by changes in the horizontal pressure gradient due to the varying density of Adriatic Deep Water (AdDW), which spreads into the deep layers of the Northern Ionian Sea. In turn, the AdDW density fluctuates according to the circulation of the NIG through a feedback mechanism known as the bimodal oscillating system. We set up laboratory experiments with a two-layer ambient fluid in a circular rotating tank, where densities of 1000 and 1015 kg m$^{-3}$ characterize the upper and lower layers, respectively. From the potential vorticity evolution during the dense water outflow from a marginal sea, we analyze the response of the open-sea circulation to the along-slope dense water flow. In addition, we show some features of the cyclonic/anticyclonic eddies that form in the upper layer over the slope area. We illustrate the outcome of the experiments of varying density and varying discharge rates associated with dense water injection. When the density is high, 1020 kg m$^{-3}$, and the discharge is large, the kinetic energy of the mean flow is stronger than the eddy kinetic energy. On the other hand, when the density is lower, 1010 kg m$^{-3}$, and the discharge is reduced, vortices are more energetic than the mean flow; that is, the eddy kinetic energy is larger than the kinetic energy of the mean flow. In general, over the slope, following the onset of dense water injection, the cyclonic vorticity associated

with current shear develops in the upper layer. The vorticity behaves in a two-layer fashion, thus becoming anticyclonic in the lower layer of the slope area. Concurrently, over the deep flat-bottom portion of the basin, a large-scale anticyclonic gyre forms in the upper layer extending partly toward a sloping rim. The density record shows the rise of the pycnocline due to the dense water sinking toward the flat-bottom portion of the tank. We show that the rate of increase in the anticyclonic potential vorticity is proportional to the rate of the rise of the interface, namely, to the rate of decrease in the upper layer thickness (i.e., the upper layer squeezing). The comparison of laboratory experiments with the Ionian Sea is made for a situation when the sudden switch from cyclonic to anticyclonic basin-wide circulation took place following extremely dense Adriatic water overflow after the harsh winter in 2012. We show how similar the temporal evolution and the vertical structure are in both laboratory and oceanic conditions. The demonstrated similarity further supports the assertion that the wind-stress curl over the Ionian Sea is not of paramount importance in generating basin-wide circulation inversions compared to the internal forcing.

## 1. Introduction

The effect of dense water outflow from marginal seas on ocean circulation has attracted great attention because it represents an important component of global thermohaline circulation (Jungclaus and Backhaus, 1994; Dickson, 1995). Numerical and laboratory studies of this phenomenon have been inspired primarily by observations of mesoscale eddies over dense water outflows in the ocean (e.g., Denmark Strait and Gibraltar Strait) (Mory et al., 1987, Whitehead et al., 1990; Lane-Serff and Baines, 1998; Lane-Serff and Baines, 2000; Etling et al., 2000). The coupling between upper layer circulation and dense water plumes has also been addressed by numerical modeling (i.e., Spall and Price, 1998), which confirmed the formation of eddies in the upper layer. The eddies were predicted to travel along isobaths with a characteristic speed that depends on reduced gravity, bottom slope, and Coriolis parameter (Nof, 1983). Early hypotheses stated that cyclones form by the stretching of the high potential vorticity water column (Spall and Price, 1998). In addition to the formation of cyclonic eddies in the lighter upper part of the water column, according to Lane-Serff and Baines (2000), secondary anticyclonic motion occupying a major part of the tank develops. This is the only mention in the literature of this type of consequence of dense water cascading down a slope.

The Ionian Sea, the deepest basin in the Mediterranean (maximum depth over 5000 m), together with its two adjacent basins, the Adriatic and Aegean Seas, represents a key area for both the eastern and western Mediterranean. It is crossed by the main Mediterranean water masses (Levantine Intermediate Water – LIW and Atlantic Water – AW), and it comprises the site of the Eastern Mediterranean Deep Water (EMDW) formation, a process that takes place mainly in the Adriatic Sea. Adriatic Deep Water (AdDW) overflows the Otranto Sill, represents the main component of the EMDW and spreads along the western continental slope as a bottom-arrested current affecting the northern Ionian circulation. Only occasionally very dense water forms in the Aegean Sea, as occurred in the early 1990s during the Eastern Mediterranean transient (EMT) (Roether et al., 1996; Klein et al., 1999). It was shown that dense Aegean water overflow affected the upper layer circulation, increasing the cyclonic vorticity in the continental slope area (Menna et al., 2019).

Analysis of long-term altimetric data reveals that the sea surface circulation in the Ionian shows peculiar characteristics (Vigo et al., 2005): at decadal time scales, it switches from a cyclonic basin-wide gyre occupying the entire northern area to anticyclonic meandering. This fact contributes to determining the thermohaline properties of the interior Ionian basin, of the

Adriatic Sea, and of the Levantine and even the Western Mediterranean basins (Gačić et al., 2013). During the cyclonic circulation mode, Ionian and Adriatic Seas are invaded by highly saline Levantine water. On the other hand, during anticyclonic circulation, the two basins are affected by low-salinity waters of Atlantic and Western Mediterranean origin (Brandt et al., 1999). For more than ten years, decadal inversions of the northern Ionian circulation have been the focus of Mediterranean scientists' attention because the phenomenon is very prominent and involves a large part of the water column (approximately 2000 m deep). There has been a long discussion about the mechanism generating such inversions, and some scientists have suggested, mainly based on numerical modeling studies, that the phenomenon is linked to the wind stress curl (see, e.g., Pinardi et al., 2015; Nagy et al., 2019). Other studies, however, showed that the wind curl variations are not strong enough to generate such changes; these studies sustain that the inversions are due to the interplay between the dense water flow (Adriatic or Aegean origin) and the Ionian horizontal circulation (e.g., Gačić et al. 2010; 2011; 2013; Theocharis et al., 2014; Velaoras et al., 2014). The long-term density variability of the bottom water associated with the salinity variations in the deep-water formation site induces reversals of the horizontal pressure gradient in the Ionian Sea and hence of the circulation pattern (Borzelli et al., 2009). The mechanism was named the Adriatic-Ionian bimodal oscillating system (BiOS) and described for the first time by Gačić et al. (2010). The purpose of this paper is to study in more detail the inversions of the open-ocean residual circulation generated by dense water flow over a sloping bottom and to understand whether this flow is strong enough to produce inversions in the upper-layer circulation similar to those observed in the Ionian Sea.

To address the impact of dense water flow on basin-wide open-sea circulation and on the formation of anticyclonic vorticity in the upper layer, we base our study on the analysis of the results of a series of rotating tank experiments with injection of dense water in the slope area. Our attention is concentrated on a two-layer system, which approximates the Ionian Sea conditions rather well. Then, we examine the response of the central abyssal plain of the idealized basin to dense water sinking and to its along-slope flow and compare the findings with observation studies. We also discuss mesoscale eddies and their specific features as a function of the dense water outflow rate. Dense water flow is quite often a time-limited phenomenon with a duration of several months after winter convection in a marginal sea; therefore, our experiments are designed to mimic this kind of condition. More specifically, we address the response time of the residual current field in the open sea area to the dense water flow with a limited duration over the continental slope. Hence, we discuss the effect of different discharge rates of dense water at the slope on the surface circulation in the open sea in an attempt to reproduce circulation inversions in the northern Ionian Sea. Rubino et al. (2020), by comparing the results of experiments in the rotating tank with those obtained by a numerical model and altimetry data, qualitatively show that the circulation inversion in the Ionian Sea can be solely explained in terms of the onset of dense water injection over the slope area. Starting from this finding, the present work goals are 1) to study the evolution of potential vorticity fields both on the slope and in the central area of the rotating tank, using the outputs of three different experiments, and 2) to compare them with vorticity obtained from altimetry (surface layer) and model-derived (deep layer) flow in the real Ionian Sea.

In the slope area, the injected saline solution creates a gravity current that quickly reaches geostrophic equilibrium due to the particular injection method employed. The gravity current structure consists of two parts: the first part takes the shape of a vein, with a mainly along-slope velocity component, and the second part takes the form of a viscous bottom layer, also called Ekman leakage, with a mainly downslope velocity component. A detailed description of the structure of a rotating gravity current is given in Wirth (2009) and Cenedese et al. (2004).

The paper is organized as follows: in section 2, we present the experimental setup, an overview of the selected experiments and data analysis methods. In section 3, measurements and results are described, while section 4 is dedicated to the comparison of the laboratory results and Ionian Sea example. Finally, section 5 presents a summary and the conclusions of the paper.

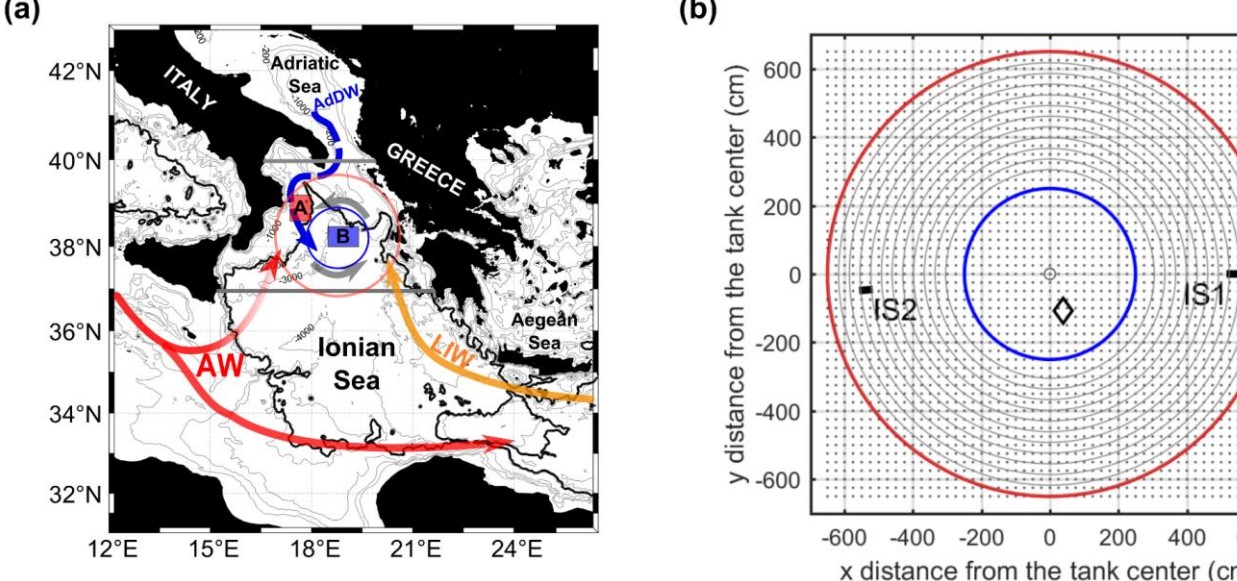

**Figure 1: (a) Map of the study area in the Ionian basin with a simplified circulation scheme, which changes according to the BiOS regime. Gray horizontal lines indicate the geographical limits within which the mean vorticities above and below the 2200 m isobath were calculated. Rectangles A and B indicate the areas where density data (CMEMS reanalysis) were averaged. Concentric rings represent the simplified laboratory tank scheme. Acronyms: AW = Atlantic Water, LIW = Levantine Intermediate Water, AdDW = Adriatic Deep Water. (b) Top view of the tank: the slope area is between the red and blue circles, the deep flat-bottom area is inside the blue ring. Dense water injectors are placed at IS1 and IS2. A diamond near the center shows the location of the Cp3 profiler. Concentric gray rings indicate intersections of the laser sheet levels with the slope. Gray dots indicate regular x-y grid nodes for the tank velocity field (subsampled every 5 nodes for clarity). The map in (a) was created from the bathymetry data ETOPO2v2, NOAA, World Data Service for Geophysics, Boulder, June 2006, doi: 10.7289/V5J1012Q) using the MATLAB software.**

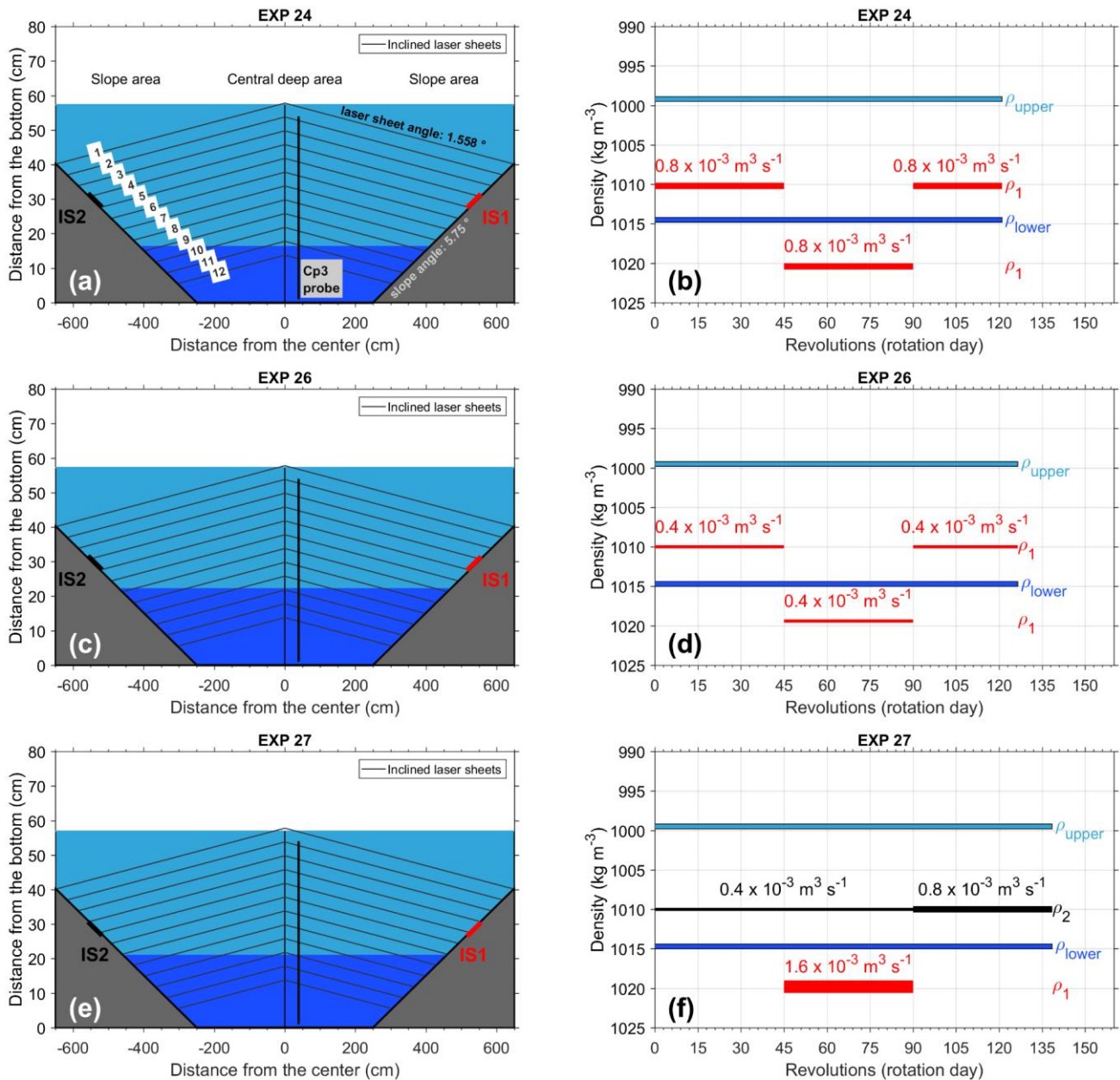

**Figure 2: Scheme of the rotating tank (not to scale) and density configuration for the three experiments discussed in the paper, EXP 24, EXP 26, and EXP 27.** Cross sectional view with central deep and slope areas, and injectors IS1 and IS2 (a, c, e); a vertical bar near the center of the tank indicates location of the vertical profiler Cp3; blue/cyan patches refer to the lower/upper layers; numbers from 1 to 12 indicate inclined laser sheet levels. Initial density (blue/cyan bars) in the lower/upper layers along with density and discharge rates of the injected water (b, d, e); red/black bars correspond to IS1/IS2; the thickness of the bars corresponds to discharge rates during various phases (for details see Tables 1 and 2). Only in EXP 27 were both injectors active.

120

## 2. Data and methods

### 2.1 Experimental design

Experiments were conducted with the Coriolis rotating platform at LEGI (Laboratoire des Écoulements Géophysiques et Industriels), Grenoble, France. A circular tank 13 m in diameter and 1 m in depth was adapted to simulate the sloping and deep flat-bottom ocean geometry (Figs. 1 and 2).

The apparatus, experimental setup, velocity measurement methods at 12 levels within the tank, their horizontal resolution, and conductivity measurements (indicators of the water density) are described in detail by Rubino et al., 2020. Here we report some essential information only. Sequences of the images of the 12 levels were taken with a high-resolution Nikon camera synchronized with a profiling laser system. It illuminated the Polyamide particles (Orgasol) with a mean diameter of 60 µm and a density of 1020 kg m$^{-3}$ dispersed in the tank and in the injected saline solutions to allow optical velocity measurements.Velocity fields were computed from the images using a cross-correlation particle image velocimetry (PIV) algorithm encoded with the software UVMAT developed at LEGI.

We distinguish the slope and central deep (flat bottom) areas in the tank that are equivalent to the continental slope and deep zone of the northern Ionian basin, respectively (Fig. 1). We compare the potential vorticity evolution in each area as related to the dense water flow. The two areas are presumably controlled by different processes of vorticity generation. In the central area (flat bottom), the upper layer squeezing, due to the downslope sinking of the dense water to the lower layer, generates the upper layer anticyclonic vorticity. In the slope area, the upper layer stretching due to the downslope water flow results in the generation of cyclonic vorticity. The lower layer on the slope is subject to squeezing and anticyclonic vorticity generation related to the formation of dense water flow parallel to isobaths.

The linear barotropic vorticity equation for an *f*-plane approximation in radial coordinates for the surface layer without wind-stress forcing is:

$$\frac{\partial \xi}{\partial t} + \frac{f}{H-h}(v * s + \frac{\partial h}{\partial t}) = 0 \qquad (1)$$

where $\xi$ is the surface layer relative vorticity, $f$ is the planetary vorticity, $h$ and $H$ are the lower layer and total fluid depth, respectively, and $s$ is the bottom slope. The radial velocity component $v$ is defined as negative if directed toward the center of the tank. In the central deep area, the second term of Eq. (1) is zero (the bottom is flat). In this study, we explore to what extent the linear vorticity equation represents a good approximation of the flow in the slope and in the central flat-bottom areas. We discuss mechanisms that should be taken into consideration in the case this equation does not satisfactorily describe the flow.

The slope area was realized using an axisymmetric conical-shaped boundary descending toward the center of the tank with a constant slope, which was determined by keeping the dynamic similarity between the phenomenon to be simulated in the rotating tank and that occurring in the Ionian Sea. The similarity results from the ratio between the gravitational (*g's*) and Coriolis accelerations (*fu*) being of the same order of magnitude for the two basins. Here, $g' = g\Delta\rho/\rho$ is the reduced gravity, $\Delta\rho$ is the density difference between the injected water and the ambient water densities, $f$ is the Coriolis parameter and $u$ is the velocity component tangential to isobaths. We introduce typical values for $g'$, $f$ and $u$ for the Ionian Sea ($g' = 1.5 \times 10^{-3}$ m s$^{-2}$, $f = 10^{-4}$ s$^{-1}$, $u = 10^{-2}$ m s$^{-}$

[1]) and take the value 5 x $10^{-2}$ for the western Ionian continental slope from the literature (Ceramicola et al., 2014). To obtain the similarity between the laboratory experiments and the Ionian Sea, we set the slope $s$ in the tank to 0.1 and $g'$ to 0.1 m s$^{-2}$; for $f$, we take 0.1 s$^{-1}$, corresponding to one rotation day (1 revolution) lasting 120 s, while for a typical speed, we chose $10^{-2}$ m s$^{-1}$. In this way, the rotating tank slope angle is equal to 5.7 $^0$, and the ratios of the two acceleration terms are of the same order of magnitude for both basins. Hence, the slope area, with a total width of 4 m, gradually descends from the tank edge, which has a height of 40 cm, down to the tank bottom. The central deep area with constant depth has a diameter of 5 m. As far as turbulent diffusivity is concerned, the comparison between the laboratory and the oceanic conditions is discussed in Appendix A.

**Table 1**. Initial conditions of the two layers and experiment duration: density ($\rho_{upper}$ and $\rho_{lower}$) and thickness ($h_{upper}$ and $h_{lower}$). Note: one revolution takes 120 s, i.e., one rotation day. $H$ is the total water thickness.

|  | $\rho_{upper}$ | $\rho_{lower}$ | $h_{upper} = H\text{-}h$ | $h_{lower} = h$ | duration |
|---|---|---|---|---|---|
|  | [kg m$^{-3}$] | [kg m$^{-3}$] | [m] | [m] | [rotation day] |
| EXP 24 | 999.2 | 1014.5 | 0.411 | 0.162 | 121.0 |
| EXP 26 | 999.5 | 1013.5 | 0.351 | 0.221 | 126.5 |
| EXP 27 | 999.5 | 1014.7 | 0.360 | 0.209 | 138.5 |

In this paper, we consider three experiments where the ambient fluid was a two-layer system (see Tables 1 and 2 and Fig. 2 for their characteristics). The upper layer was always freshwater, while the lower layer had a density of 1015 kg m$^{-3}$. Dense water was injected from a single injector (IS1) or from the pair of injectors (IS1 and IS2). The latter case simulated an EMT event when presumably two dense water sources (Adriatic and Aegean Seas) were active. In the first part of all experiments (phase I, until the 45th day), water of intermediate density (1010 kg m$^{-3}$) was injected, and then, to mimic the dense-water discharge event, water of high density (1020 kg m$^{-3}$) was injected for another 45 days (phase II). Finally, experiments ended with an approximately 30-day interval of intermediate-density water injection (phase III). The discharge rate was varied with the aim of studying its influence on the pattern of both open-sea and continental slope residual circulations, on the vorticity field and on eddy formation.

**Table 2**. Configuration of injection sources IS1/IS2: $\rho_1/\rho_2$ and $Q_1/Q_2$ are the densities and discharge rates, respectively, of injected water.

| | IS1 | | IS2 | | |
|---|---|---|---|---|---|
| | $\rho_1$ | $Q_1$ | $\rho_2$ | $Q_2$ | duration/phase |
| | [kg m$^{-3}$] | [10$^{-3}$ m$^3$ s$^{-1}$] | [kg m$^{-3}$] | [10$^{-3}$ m$^3$ s$^{-1}$] | [rotation day] |
| EXP 24 | 1010.2 | 0.8 | | | 0-45 (phase I) |
| | 1020.4 | 0.8 | | | 45-90 (phase II) |
| | 1010.2 | 0.8 | | | 90-121 (phase III) |
| EXP 26 | 1010.4 | 0.4 | | | 0-45 (phase I) |
| | 1019.4 | 0.4 | | | 45-90 (phase II) |
| | 1010.4 | 0.4 | | | 90-126.5 (phase III) |
| EXP 27 | - | - | 1010.4 | 0.4 | 0-45 (phase I) |
| | 1019.8 | 1.6 | 1010.4 | 0.4 | 45-90 (phase II) |
| | - | - | 1010.4 | 0.8 | 90-138.5 (phase III) |

## 2.2 Data analysis

From the eastward and northward current components, we calculated radial and tangential velocities (system in polar coordinates); the radial velocity ($v$) is defined as positive from the tank center outwards, and the tangential one ($u$) is perpendicular to $v$ and positive in a counterclockwise sense.

        The horizontal spatial resolution for all velocity components, as well as for other derived fields, such as vorticity, at each of the 12 vertical levels in Fig. 2 is 5 x 5 cm.

For each vertical profile of density measured in the central deep area (by means of the probe Cp3, Fig. 2), we calculated the mixed layer depth (MLD) using the threshold method with a finite difference criterion (de Boyer Montegut et al., 2004), considering a 1.5 kg m$^{-3}$ vertical density gradient threshold and, as a density reference, the value at 5 cm depth. On the other hand, we determined the base of the pycnocline using the same method but taking as the density reference the values at 56, 50, 50 cm depth for EXP 24, EXP 26 and EXP 27, respectively.

Regarding the terms of the quasigeostrophic linear vorticity equation (Eq. 1), the surface is defined as the mean of levels 2 to 5, and the bottom is defined as the mean of levels 10 and 11. Level 1 is discarded because in the central deep area, it is too close

to the free surface and is therefore noisy. In the computation of the derivative of the curl, both the vorticity time series and its derivative are smoothed with a moving average of 7 points.

In the calculation of the time-lag cross correlation of vorticity over the tank slope area, we averaged vorticity values over levels 1 to 4 in each of three arbitrarily chosen zones. In the slope area, levels 1-4 are well inside the upper layer of the tank. The time series are lag-shifted for maximum positive and negative time lags imposed by the integral time scale, i.e., the sum of the normalized autocorrelation function, which gives the measure of the dominant correlation time scale (Emery and Thomson, 2001) and is calculated for the entire slope area of the tank.

Mean Kinetic Energy (MKE) and Eddy Kinetic Energy (EKE) were computed in the surface layer over the slope based on the time series of current velocity components $v_x$ and $v_y$ according to the system in tank coordinates (Fig. 1b). More specifically, we took $v_x$ and $v_y$ as the respective average from the levels 1, 2, 3, and 4 at each grid point. Hence, MKE $= \frac{1}{2}\,(<v_x>^2 + <v_y>^2)$ and EKE $= \frac{1}{2}\,(<v_x'^2> + <v_y'^2>)$, where the symbol $<>$ means the temporal average in each grid point, $v_x' = v_x - <v_x>$, and $v_y' = v_y - <v_y>$. This operation was performed for each measurement phase for which, finally, spatial averages of MKE and EKE were obtained over the slope area.

We compare the current fields in the rotating tank and in the real ocean for a particular condition when a circulation inversion event was observed in the northern Ionian Sea. Regarding the real ocean, for the surface we use the altimetry data, while for the deep layer conditions we use outputs from the hydrodynamic model of the Mediterranean Forecasting System. The latter concerns the physical reanalysis component, originating from the Copernicus Marine Service MEDSEA_REANALYSIS_PHYS_006_004 dataset (CMEMS Reanalysis) supplied by the Nucleus for European Modelling of the Ocean (NEMO) (Simoncelli et al., 2019). The model has a horizontal grid resolution equal to $1/16°$ (ca. 6-7 km) and 72 unevenly spaced vertical levels. We use the following variables: 3D monthly mean and daily mean temperature, salinity, and horizontal current components (eastward and northward) covering the entire Mediterranean Sea (https://resources.marine.copernicus.eu/?option=com_csw&view=details&product_id=MEDSEA_MULTIYEAR_PHY_006_00 4). Daily absolute geostrophic velocities (AGVs) derived from altimetric (surface) and CMEMS Reanalysis (1000 m depth) current data were used to estimate the relative vorticity. The resulting geostrophic vorticity fields were spatially averaged in the northern Ionian Sea (37°-40°N; 15°-21°E), separately for the central area (bins located at depths greater than 2200 m) and for the slope (bins located at depths less than 2200 m). Time series of these spatially averaged parameters were normalized (divided by the Coriolis parameter) and filtered using a 61-day moving average.

### 3. Results of data analyses

### 3.1 Density in the central deep area

The dense fluid sinks from the source toward the central portion of the tank within the distance on the order of one radius of deformation and then turns and flows along the isobaths (Smith, 1975; Juncklaus and Backhaus, 1994; Lane-Serff and Baines, 1998). Due to the sinking of the dense fluid, the interface, intended as the upper boundary of the pycnocline layer, in the central deep area of the tank rises. The rate of the interface rise is proportional to the volume discharge rate assuming that this water is distributed evenly over the entire basin:

$$\frac{\partial h}{\partial t} = \frac{Q}{\pi R(t)^2},$$
(2)

where $R(t) = R_0 + \left(\frac{h}{s}\right)$ is the radius of the cylinder filled by the discharge and $R_0$ is the radius of the interface between the upper and lower layers at $t = 0$.

The estimate of the $\partial h/\partial t$ can be compared with experimental data of lower layer thickness variations, i.e., the interface movements, obtained from the density profiling near the center of the tank at Cp3. The two values should coincide if the dense water volume is evenly distributed over the entire tank. The interface rise then results in the squeezing of the upper layer, which is proportional to the rate of increase of the anticyclonic vorticity of the surface layer (see Eq. 1) in the central part of the basin. In contrast, in the surface layer in the slope area, we should expect the generation of the cyclonic vorticity associated with the downslope flow of dense water. This dense current carries overlying ambient fluid to deeper areas, causing stretching of the water column. Conversely, the bottom layer in the slope area should be squeezed due to the downslope flow of dense water.

Vertical density profiling during experiments enables us to reconstruct either the MLD or the lower layer thickness (*h*) as a function of time. We consider the lower layer thickness from experimental data to be the total water depth in the tank minus the depth of the MLD (cyan solid line in Fig. 3). The rise of the interface associated with the dense water sinking down the slope to the lower layer of the central deep area is clear from the Hovmöller diagram. At the beginning of the experiment, the halocline is very thin but thickens with time due to eddy diffusion. Lower layer thickness shows different rates of change for different experiments. The rate of change, as follows from Eq. (2), is directly proportional to the discharge rate and inversely proportional to the area of the base of the volume occupied by the injected water.

Integrating Eq. (2) we obtain:

$$h = sR_0 \left\{ \left[ \left(1 + \frac{h_0}{sR_0}\right)^3 + \frac{3Qt}{\pi sR_0^3} \right]^{\frac{1}{3}} - 1 \right\}$$
(3)

where the subscript 0 indicates quantities at $t = 0$.

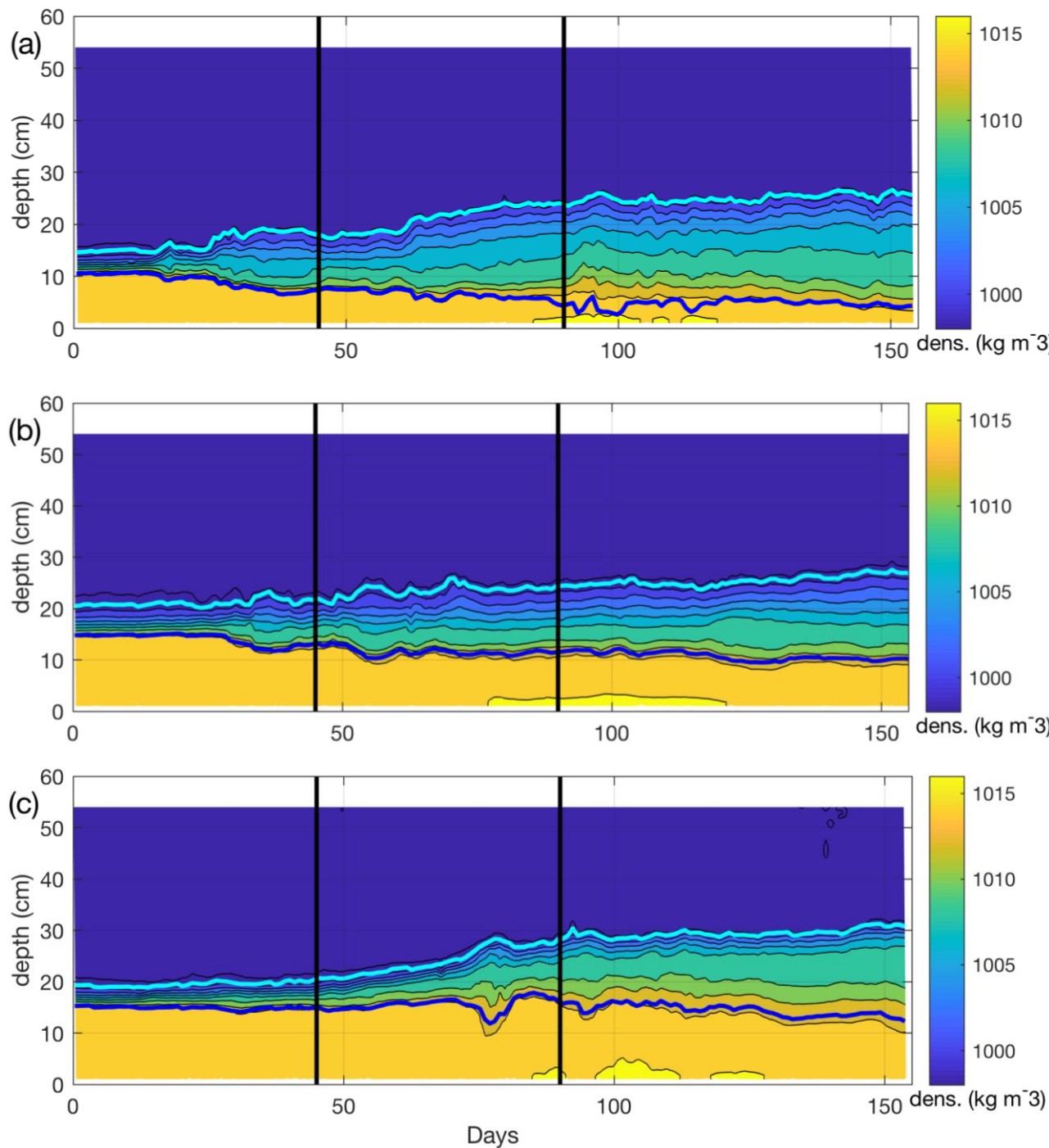

**Figure 3: Temporal evolution of density and the mixed layer depth (MLD, thick cyan line) for experiments 24 (a), 26 (b) and 27 (c). The base of the pycnocline is denoted by the thick blue line. Black solid lines indicate rotation days 45 and 90, i.e., the start and end of the high-density water injection, respectively (for reference see Fig. 2 and Tables 1 and 2). The black isoline interval is 2 kg m$^{-3}$, starting from 1016 kg m$^{-3}$ at the bottom.**

The slowest decrease in the upper layer thickness, i.e., an MLD decrease, is evident during experiment 26, which is characterized by the lowest dense water discharge rate. In that case, the rise (descent) of the upper (lower) boundaries of the

pycnocline layer probably depends both on dense water sinking and vertical mixing, which provokes thickening of the pycnocline layer. The temporal evolution of the MLD for the other two experiments is mainly due to dense water sinking, while vertical mixing and turbulent diffusion probably play a minor role. This is evident from the rise of the interface (that is, the upper boundary of the pycnocline) being larger than the deepening of the lower boundary. Indeed, as already pointed out, the rate of the interface rise is directly proportional to the discharge rate and inversely proportional to the square of the radius of the base of the volume occupied by the discharged water as a function of time (Eq. 2). We limit our study to cases in which the filling up of the lower layer with dense water is to a larger extent responsible for the interface rise, i.e., to experiments 24 and 27.

We compare the MLD temporal evolution with variations in the lower layer thickness from the theoretical relationship (Eq. 3). The interface rise as a function of the dense water injection rate (Fig. 4) is a good approximation of the interface depth variations in the central deep area. In fact, the slopes of the curves are remarkably similar. The offset between the theoretical curve and the experimental data is probably present because the interface is not a plane but a layer of finite thickness due to vertical diffusion. The anomaly around rotation day 75$^{th}$, which is clearly seen from the experimental curve of lower layer thickness for experiment 27 (Fig. 4b) and also evident as a temporary thickening of the interface layer (Fig. 3c), is discussed more thoroughly in Appendix B.

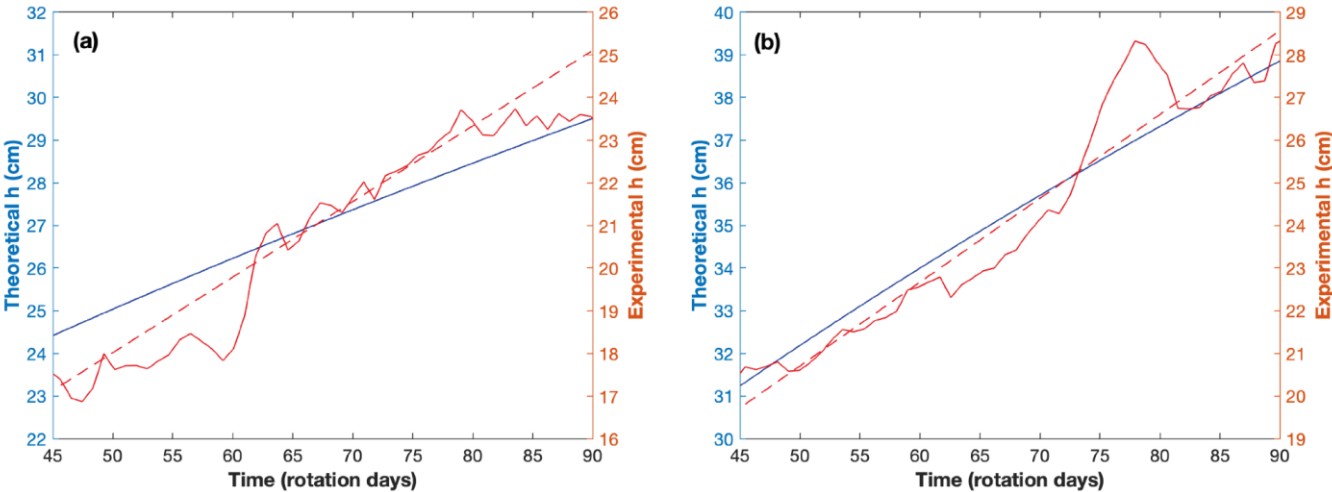

**Figure 4: Evolution of the lower layer thickness during phases I and IIs: experimental data from vertical profiling at Cp3 (continuous red line), linear regression curve (dashed line) and the theoretical values obtained from Eq. 4 (see text, blue line). EXP 24 in (a) and EXP 27 in (b).**

### 3.2 Vorticity in the upper and lower layers

The evolution of the potential vorticity of the upper layer for the central deep area is directly proportional to the dense water injection rate (Eq. 1). Here, we assume that the interface is horizontal over the entire central deep area. Dividing Eq. (1) by $f$ and defining $\zeta$ as a relative vorticity normalized by the planetary vorticity $f$ ($\zeta = \xi/f$), and considering Eq. (2), we obtain the quasigeostrophic potential vorticity equation for the flat bottom:

$$\frac{\partial \zeta}{\partial t} + \frac{1}{H-h} \frac{\partial h}{\partial t} = 0, \tag{4}$$

or from Eq. (2) we have:

$$\frac{\partial \zeta}{\partial t} + \frac{1}{H-h} \frac{Q}{\pi R^2} = 0, \tag{5}$$

      The temporal evolution of the vorticity in the two presented experiments (Fig. 5) indeed confirms the above considerations. In the case of experiment 24, when the dense water discharge rate is set to a constant value of $0.8 \cdot 10^{-3}$ m$^3$ s$^{-1}$, the vorticity in the upper layer of the central deep area begins to decrease immediately after the start of the experiment, becoming anticyclonic after 20 rotation days. It continues to decrease gradually until the end of phase II. The slope of the vorticity curve does not change

appreciably despite the high-density water discharge (1020 kg m$^{-3}$) during phase II. At the same time, the vorticity in the upper layer of the slope area shows a monotonic increase, reaching smaller absolute values than the anticyclonic vorticity in the central deep area. Lower-layer vorticity evolution (Fig. 5 a and c) over the slope area shows an increase in the anticyclonic vorticity, while the vorticity in the central deep area is rather small.

      In the case of experiment 27 (Fig. 5 b and d), the evolution of the vorticity in both the upper and lower layers of the deep

central and slope areas shows features similar to those in experiment 24. Differences between the two experiments occur in the initial part of the time series, when the injection rate of water of intermediate density (1010 kg m$^{-3}$) is only $0.4 \cdot 10^{-3}$ m$^3$ s$^{-1}$ in experiment 27. In experiment 24, the injection rate starts with $0.8 \cdot 10^{-3}$ m$^3$ s$^{-1}$ and remains constant throughout the entire duration. In experiment 27, prominent vorticity variations occur only during phase II, when the high-density water (1020 kg m$^{-3}$) has a much higher discharge rate ($1.6 \cdot 10^{-3}$ m$^3$ s$^{-1}$). Approximately ten days after the onset of phase II, the vorticity suddenly changes, becoming

anticyclonic in the surface layer of the central area, and almost simultaneously, the flow becomes cyclonic in the upper layer of the slope. In the central deep area, there is no clear signal of the lower-layer vorticity variations. After the cessation of the high-density water discharge, the water of density 1010 kg m$^{-3}$ continues to be released at a discharge rate of $0.8 \cdot 10^{-3}$ m$^3$ s$^{-1}$ (phase III), which yields a decrease in the anticyclonic vorticity in the surface layer in the central deep area.

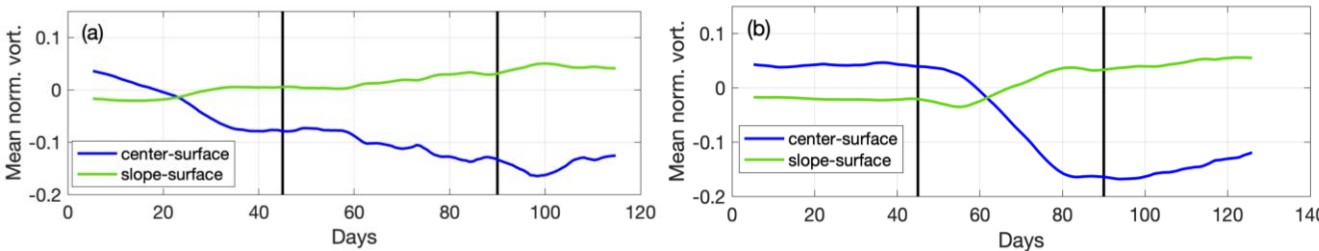

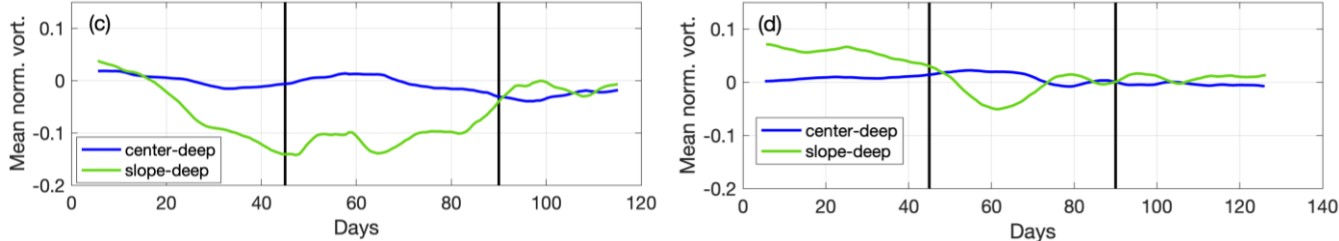

**Figure 5: Evolution of the average vorticity normalized by the Coriolis parameter for EXP24 (a, c), and EXP27 (b, d). Vorticity curves are smoothed with a moving average of 15 points. In the legends, "center" refers to the central area of the tank (upper or lower layer), while "slope" refers to the slope area (upper or lower layer).**

To summarize, in the slope area, the vorticity behaves in a two-layer fashion. In the case of the lower layer in experiment 24, it becomes anticyclonic a few days after the beginning of the experiment, while the upper layer (Fig. 5 a and c) becomes cyclonic with an approximately ten-day phase lag with respect to the lower layer. In the case of experiment 27, the generation of anticyclonic vorticity takes place only after the onset of dense water injection with a high discharge rate ($1.6 \ 10^{-3} \ m^3 \ s^{-1}$, phase II), while in phase I, we set up a very low discharge rate that generally did not affect the vorticity field. Again, the response in the upper-layer lags the lower-layer vorticity. The vertical structure in the central deep area for both experiments shows the occurrence of strong anticyclonic motion in the upper layer, while in the lower layer, there is no well-defined vorticity behavior. The two-layer distribution of the vorticity at the slope area can be explained in terms of a small vertical-to-lateral friction ratio, whereas the single-layer circulation occurring occasionally in the center of the tank is supported by a relatively large vertical-to-lateral friction ratio (Orlić and Lazar, 2009). This also suggests that we cannot neglect the frictional influence, especially at the slope.

Equation (5) suggests that for the flat bottom (central deep area of the rotating tank), the rate of change in the upper-layer normalized relative vorticity is equivalent to the inverse residence time (Monsen et al., 2002), i.e., the ratio of the dense water injection rate and the volume of the upper layer. The rate of change in $\zeta$ is also proportional to the time derivative of the lower-layer thickness (Eq. 4). Variations in the upper-layer potential vorticity (Fig. 5) qualitatively confirm the results of Eq. (5), i.e., the time derivative of the normalized potential vorticity is smaller for experiment 24 than for experiment 27. Indeed, in the case of the former experiment, the dense water discharge rate is half of the latter. This does not mean that the rate of change in the potential vorticity is twice as large for experiment 27 as for experiment 24 because it additionally depends on the upper-layer thickness and on the radius R, which are different for the two experiments.

From the flow field and the MLD, we can check how successfully the quasigeostrophic potential vorticity equation for the flat bottom approximates the surface flow characteristics. The rate of change of the normalized potential vorticity is negatively correlated with the rate of change in the lower-layer thickness (Fig. 6) as follows from Eq. (2). In fact, the calculated correlation coefficients (Table 3) are negative and statistically significant at the 95% confidence level. Therefore, in the flat-bottom area, the quasigeostrophic potential vorticity equation approximates the upper-layer vorticity evolution rather well. In addition, it is evident that curves of the rate of change of the potential vorticity and the upper-layer depth for experiment 24 are noisier than those for experiment 27, which is probably due to the more prominent mesoscale activity in the former experiment than in the latter, as will be shown later.

In the slope area, we must also consider the topographic β-term (Eq. 1). Thus, we compare the rate of the vorticity change with the sum of the rate of change in the lower-layer thickness and the topographic β-term for the two experiments (Fig. 7). As the downslope dense-water flow generates stretching of the water column in the upper layer and consequently cyclonic vorticity, we average lower-layer radial velocity over the entire slope area. In fact, for a major part of the experiments, the topographic β-term is negative (graph not shown), suggesting that cyclonic vorticity generation takes place due to the cross-isobath, downslope dense water flow. Generally, the correlation coefficients for the slope area are smaller than those for the central deep area. This confirms that at the continental slope, the quasigeostrophic vorticity equation does not describe the vorticity variations as successfully as in the flat-bottom area. This can probably be explained by the fact that the bottom viscous draining at the slope area is not included in the quasigeostrophic vorticity equation.

**Table 3. (a)** Correlation coefficients (*r*) between the terms of the vorticity equation (Eq. 1) for the central deep part of the tank. Upper and lower limits of the coefficients are given at the 95% confidence level (c.l.).

| $\frac{1}{f}\frac{\partial \xi}{\partial t}$ and $\frac{1}{H-h} * \frac{\partial h}{\partial t}$ | Entire experiment duration | | Period 45-90 rot. days (Phase II) | |
|---|---|---|---|---|
| | *r* | Limits 95% c.l. | *r* | Limits 95% c.l. |
| EXP24 | -0.5466 | -0.66/-0.41 | -0.6306 | -0.77/-0.43 |
| EXP27 | -0.6357 | -0.72/-0.53 | -0.4581 | -0.66/-0.20 |

**Table 3. (b)** Correlation coefficients (*r*) between the terms of the vorticity equation (Eq. 1) for the slope of the tank. Upper and lower limits of the coefficients are given at the 95% confidence level (c.l.).

| $\frac{1}{f}\frac{\partial \xi}{\partial t}$ and $\frac{1}{H-h}(\frac{dh}{\partial t} + v * s)$ | Entire experiment duration | | Period 45-90 rot. days (Phase II) | |
|---|---|---|---|---|
| | *r* | Limits 95% c.l. | *r* | Limits 95% c.l. |
| EXP24 | 0.4202 | 0.27/0.55 | 0.6268 | 0.42/0.77 |
| EXP27 | 0.3930 | 0.24/0.52 | 0.4552 | 0.20/0.65 |

### 3.3 Eddies over the slope area

The temporal evolution of the flow field reveals that the two chosen experiments exhibit eddy formation on the slope but not of the same intensity. Animated maps of the horizontal flow at level 1 for the entire experiment duration (animations S1 and S2 in the Supplementary Material) support this difference. Here, we illustrate eddy formation by showing two selected snapshots

from each of the three phases of the experiments (Fig. 8). Although eddies are not our primary interest, we briefly address their characteristics in experiments 24 and 27 since, as stressed by Whitehead et al. (1990), "Isolated eddies are some of the most beautiful structures in fluid mechanics". Qualitatively speaking, experiment 24 shows stronger eddy activity than experiment 27. To explain the differences between the two experiments in terms of eddy formation, we calculated the relative vortex stretching

and the relative importance of viscous draining following the approach by Lane-Serff and Baines (2000). With reference to Fig. 3 in that paper, our calculations indeed show that experiment 24 during phase I falls within the parameter space where the relationship between the relative stretching and the relative importance of viscous draining supports the formation of eddies.

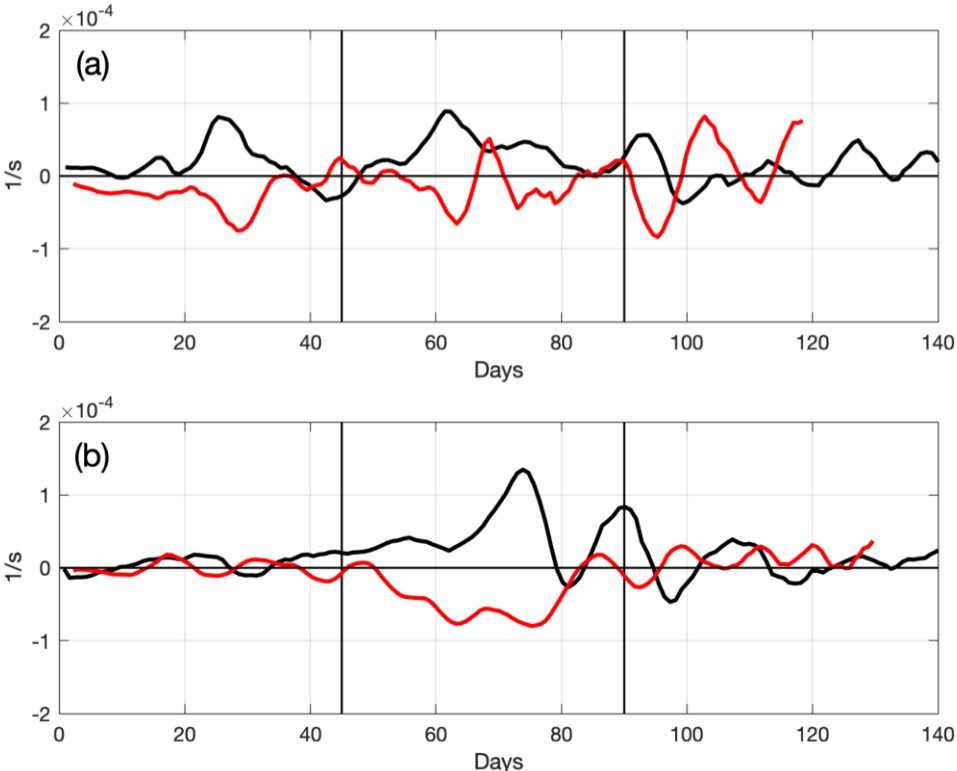

**Figure 6: Time series in the central deep area of the rotating tank (flat bottom): the vorticity rate of change ($\frac{1}{f}\frac{\partial \xi}{\partial t}$, red line) and the rate of change of the lower layer thickness ($\frac{1}{H-h}\frac{\partial h}{\partial t}$, black line) for EXP24 (a) and EXP27 (b). For reference, see Eq. 1, when slope s = 0.**

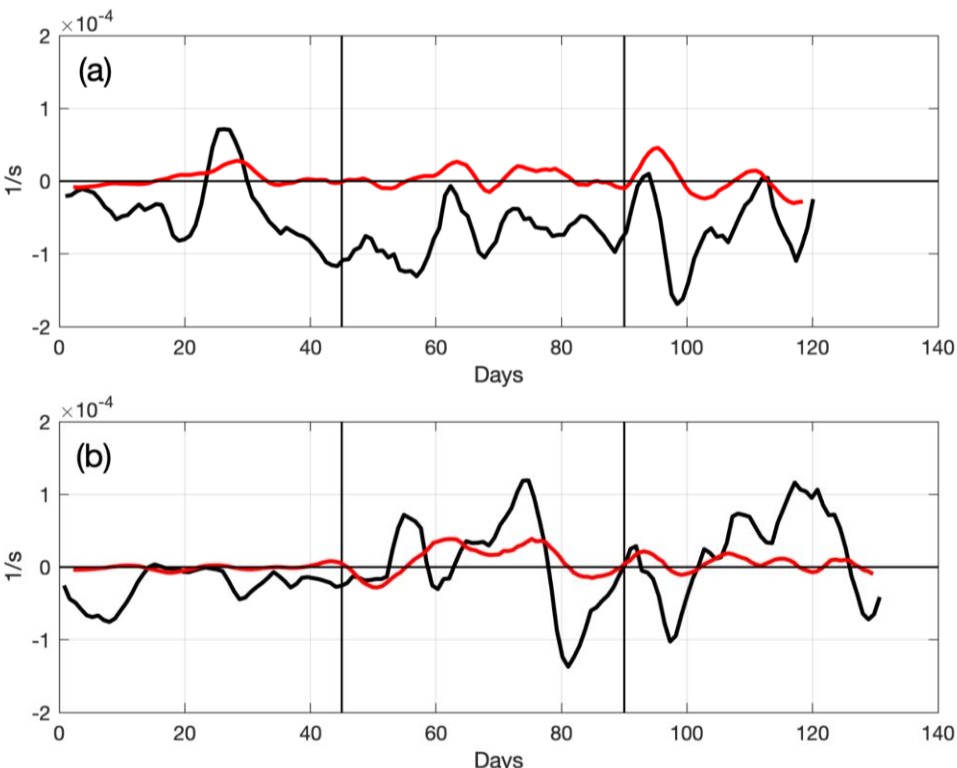

**Figure 7: Time series in the slope area: the vorticity rate of change ($\frac{1}{f}\frac{\partial \xi}{\partial t}$, red line) and the sum of the rate of change of the lower layer thickness and the topographic β-term ($\frac{1}{H-h}(\frac{\partial h}{\partial t} + v\ s)$, black line) for EXP 24 (a) and EXP 27 (b). For reference, see Eq. 1, when slope s = 0.1.**

Experiment 27, on the other hand, has a discharge rate of only 0.4 $10^{-3}$ m$^3$ s$^{-1}$ during phase I and falls within the overlapping region; thus, eddies are less likely to occur. In addition, experiments 24 and 27 during phase II both fall in the overlapping region where eddy formation is less likely to take place. During phase III, when the discharge rate is 0.8 $10^{-3}$ m$^3$ s$^{-1}$, the two experiments fall within the eddy region in the parameter space. Hence, mesoscale eddies are likely to be formed most of the time in experiment 24 but only during the last part in experiment 27.

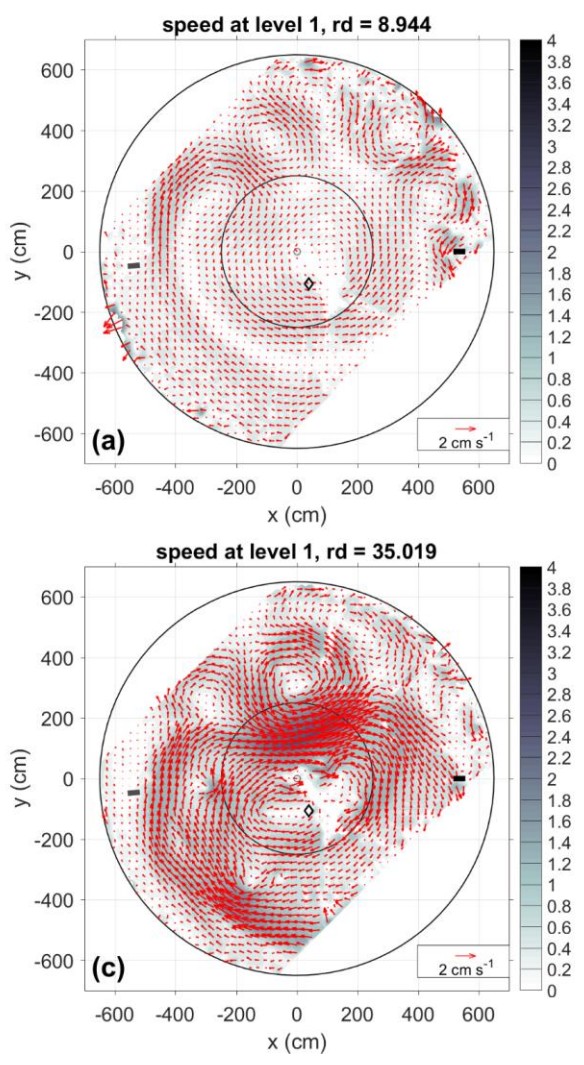
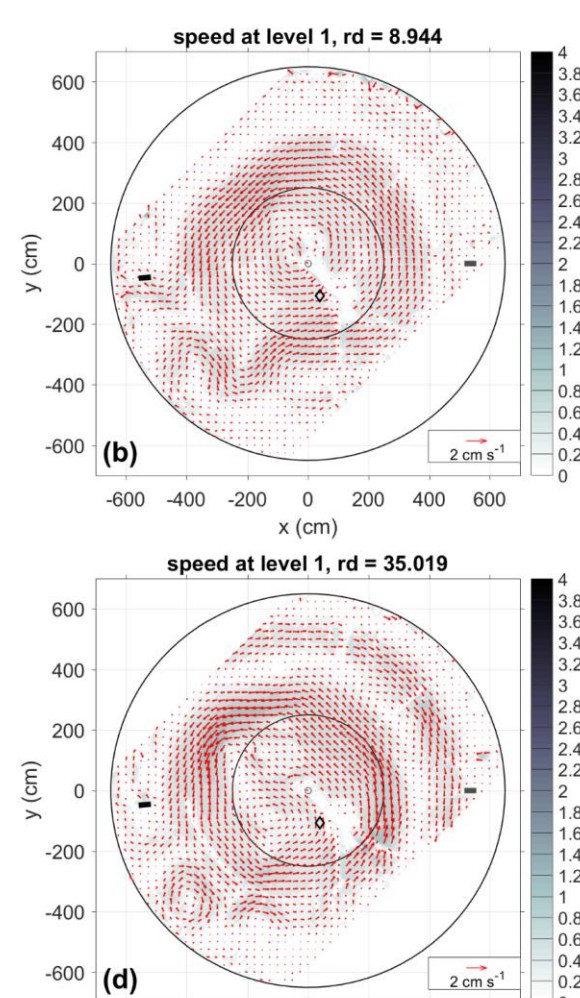

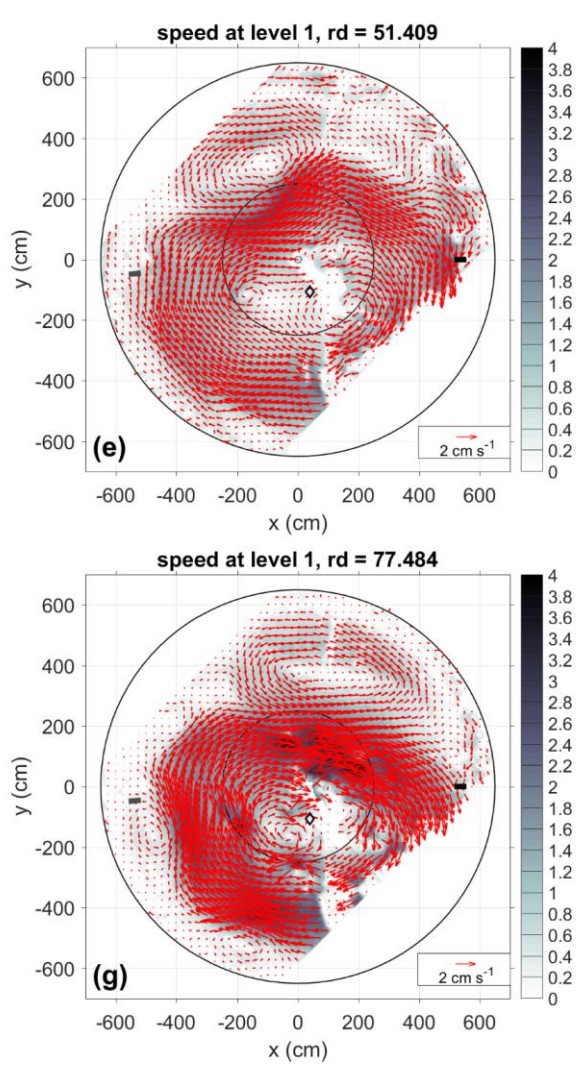
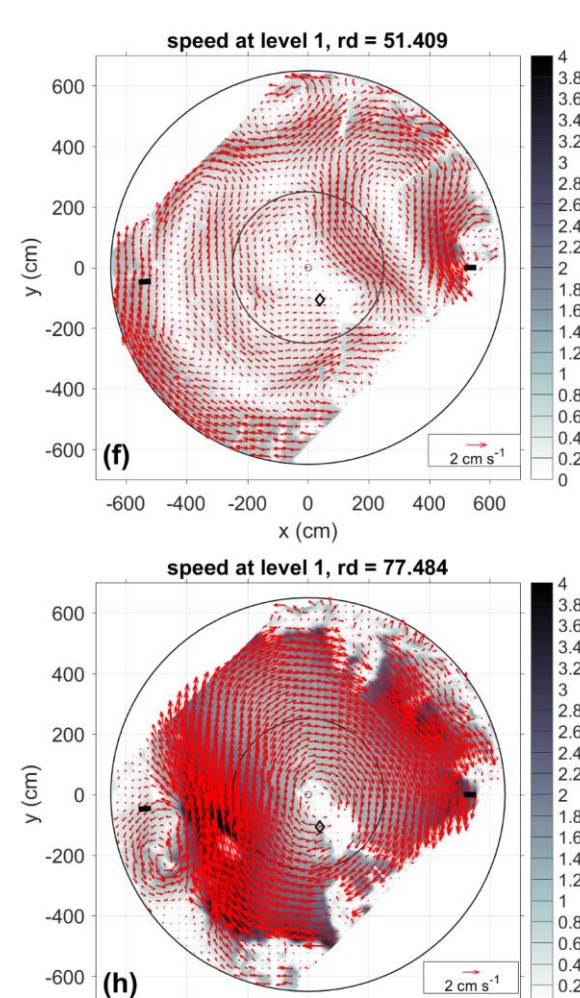

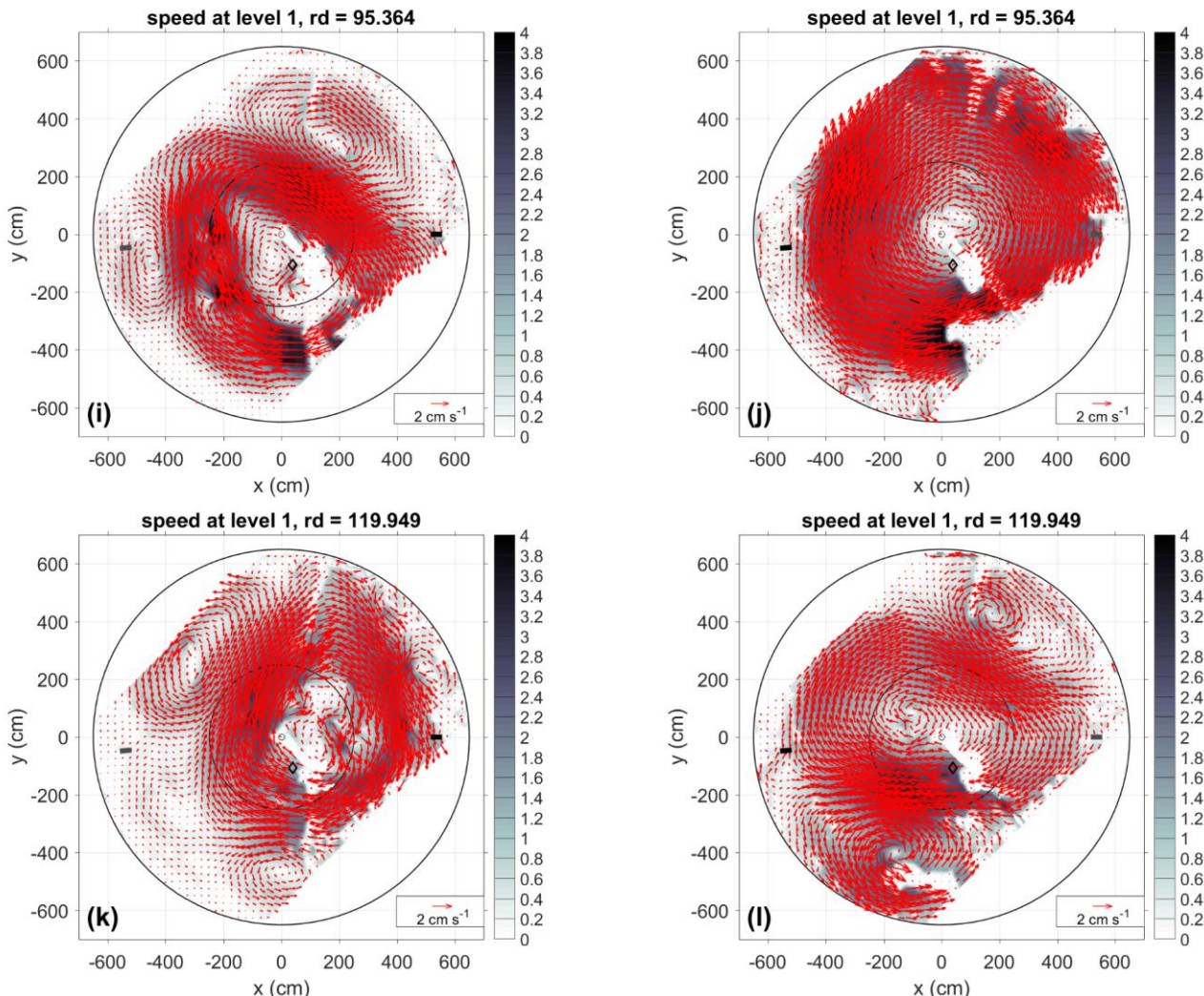

**Figure 8: Two realizations of the current field at level 1 (surface) from each phase: I, i.e., before the high-density water injection, rd <45 (a, b, c, d), II, i.e., when the high-density water injection is switched on, 45<rd<90 (e, f, g, h), and III, i.e., when the high-density water injection is switched off, rd> 90 (i, j, k, l). Experiment 24 (a, c, e, g, i, k) and experiment 27 (b, d, f, h, j, l); rd stands for rotation day.**

The ratio between the eddy kinetic energy per unit mass (EKE) and the kinetic energy of the mean flow per unit mass

(MKE) measures the relative importance of eddies with respect to the mean flow. Thus, we calculate the average EKE and MKE

for both experiments for the three phases over the entire slope area, as well as the ratio between the two (Fig. 9). It is evident that

in phase I in experiment 24, EKE > MKE, i.e., eddies are more energetic than the mean flow (MKE), as already pointed out by the

relationship between the relative stretching and the relative importance of the viscous draining. For both experiments, MKE is

larger than EKE in phase II, as follows from the fact that the two experiments fall within the overlapping region where eddies are

400 less likely to be formed. Finally, in phase III, both experiments have larger MKE values than EKE values, although they fall within

the eddy region. This can be due to the rather high remaining energy of the mean flow from phase II being larger than the energy

of the newly formed eddies.

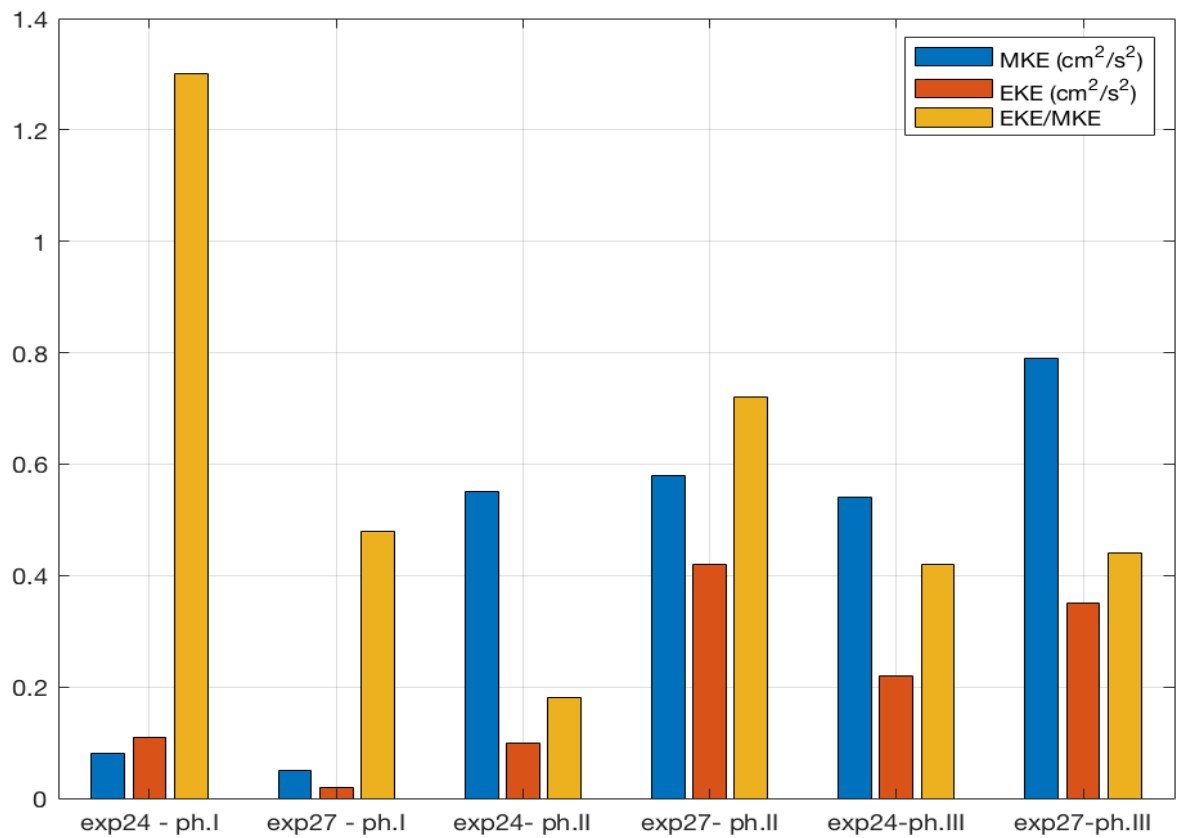

**Figure 9: Eddy kinetic energy (EKE) and kinetic energy of the mean motion (MKE) in the surface layer (levels 1 to 4), spatially averaged over the slope area during phases I, II, and III (i.e., before, during, and after the high-density water injection) for experiments 24 and 27.**

The horizontal distribution of current vectors in phase I shows dissimilarities between experiments 24 and 27. The differences are partly due to the different dense water discharge rates ($0.4 \cdot 10^{-3}$ m$^3$ s$^{-1}$ for 27 and $0.8 \cdot 10^{-3}$ m$^3$ s$^{-1}$ for 24). In both experiments, we see the formation of mesoscale eddies and their progression downstream from the source. However, in the case of experiment 27, eddies are anticyclonic and weaker than in the case of experiment 24. Downstream of the source of the dense water in experiment 24, cyclonic eddies prevail. This could be associated with the larger distance between the dense water source and the interface in experiment 24 resulting in a longer sinking path of the dense water and more intense stretching of the water column. It should be specified that the dense water entering the upper layer during phase I has a density of 1010 kg m$^{-3}$ in both experiments, while the lower-layer water density is 1015 kg m$^{-3}$; thus, the inflowing water spreads along the interface. The prevalence of anticyclonic eddies during experiment 27 is probably because the dense water source is immediately above the interface, and thus, only reduced column stretching takes place resulting in a weak cyclonic vorticity generation. Further downstream of the dense water source more pronounced presence of anticyclonic eddies is due to the upper-layer squeezing because of the alongslope flow at the interface.

During phase II, when the dense water discharge is stronger in experiment 27 than in experiment 24, toward the end of the phase, the flow field shows strong clockwise basin-wide circulation that is more spatially coherent in experiment 27 than in

experiment 24. Eddy activity as a residual of phase I is still prominent in experiment 24. Phase III with the same dense-water flow rate for both experiments shows the slowdown of the basin-wide anticyclonic circulation and the relative increase in mesoscale activity.

To study in more detail the evolution of the vorticity field in the upper layer of the slope area, we calculate the average vorticities for three chosen zones on the slope of the tank (Fig. 10a) and the lagged cross-correlation between them to estimate the eddy propagation direction and speed. As an example, we present here the results for experiment 24, which has prominent eddy activity in the slope area, as shown earlier. The cross-correlation between zones 1 and 2 (Fig. 10b) displays a significant maximum for the positive phase lag at approximately 21 rotation days, suggesting that the vorticity structures propagate prevalently anticlockwise between the two zones, i.e., leaving the shallow water on their right. From this phase lag, we calculate the propagation speed, which is approximately 0.3 cm s$^{-1}$, and this value is smaller than the eddy speed (1.3 cm s$^{-1}$) estimated in the work by Lane-Serff and Baines (1998). On the other hand, the cross-correlation between zones 1 and 3 in general is the most significant, showing a peak for a negative phase lag of approximately ten days, revealing the propagation of the vorticity structures in the opposite direction, with a speed of approximately 1.1 cm s$^{-1}$. This is probably associated with the average advection speed of the basin-wide anticyclonic flow. The change in the propagation direction can be explained by the fact that vorticity structures (eddies and meanders) first propagate cyclonically from the dense water source, as noticed in various papers and experiments (see, e.g., Etling et al., 2000 and references cited therein); then, they start to be advected by basin-wide anticyclonic flow, which develops approximately ten days after the beginning of the experiment due to surface-layer squeezing. The cross-correlation between zones 2 and 3 is the least significant, probably because zone 2 is under the combined influence of eddies moving in opposite directions. It is evident from the horizontal distribution of current vectors (see Supplementary material) that cyclonic or sometimes anticyclonic eddies continuously form downstream of the dense water source due to upper-layer stretching in the case of high-density water spreading or to upper-layer squeezing associated with intermediate density (1010 kg m$^{-3}$) water spreading along the interface between the upper and lower layers. These eddies move counterclockwise until they experience advection by anticyclonic basin-wide flow and start moving in the opposite direction.

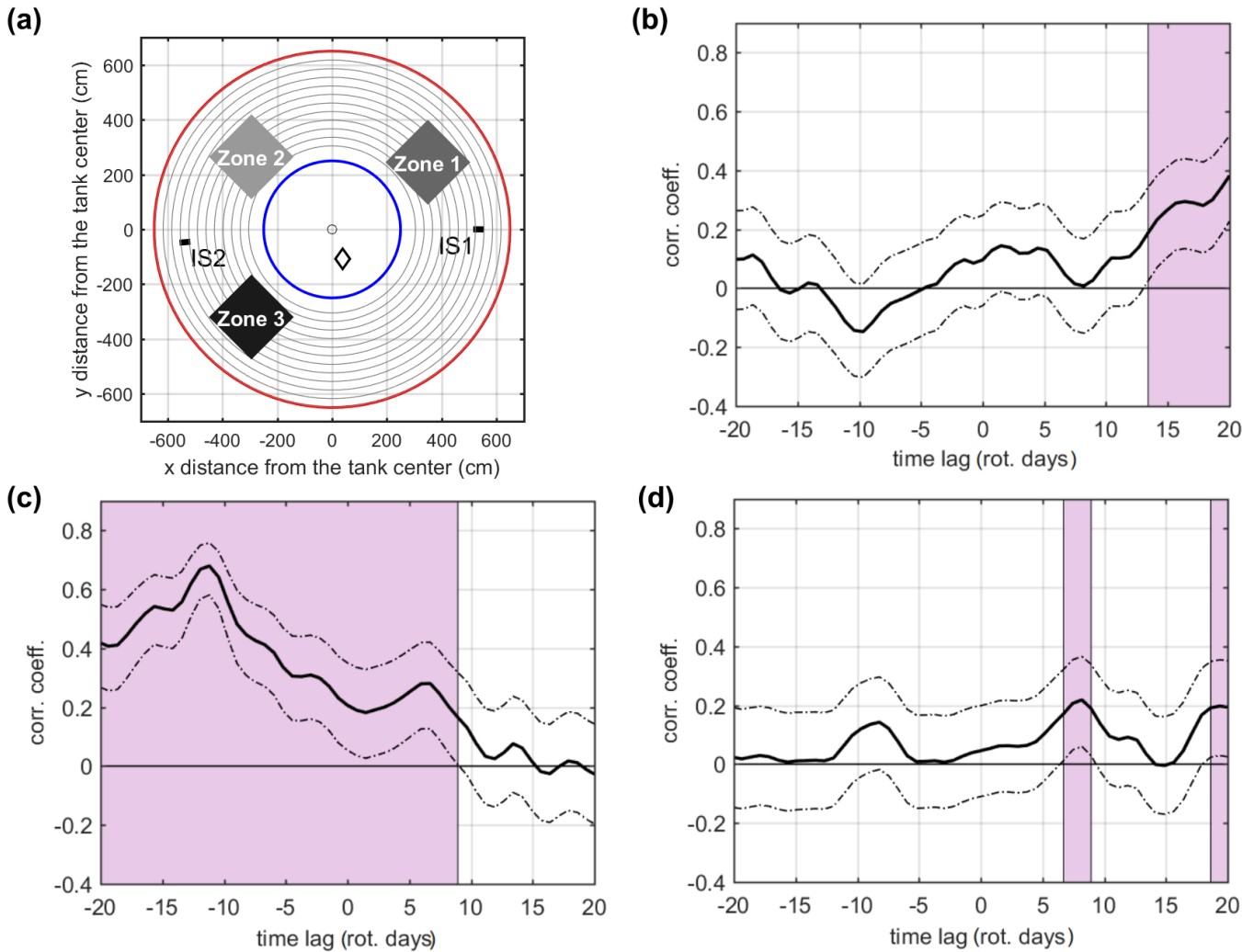

**Figure 10: Vorticity correlations over the slope for EXP24. Position of the three selected zones (a); time-lagged correlation coefficients (solid line) between zones 1 and 2 (b), between zones 1 and 3 (c); between zones 2 and 3 (d). Upper and lower limits of the coefficients, corresponding to the 95% confidence level, are indicated by dashed-dot lines. Rose shaded areas indicate a positive correlation at 95% confidence limits. The limits for the maximum time lag were set by the calculated correlation time for the entire slope area, equal to**
450 **approximately 22 rotation days.**

## 4. Comparison between the laboratory experiment and the Ionian Sea

We compare laboratory experiments, which simulate the effects of dense water overflow into the real ocean, with an event that occurred in the northern Ionian Sea characterized by a sudden change in circulation as a consequence of very dense water flowing from the Adriatic Sea following a harsh winter (Mihanović et al., 2012; Bensi et al., 2013; Raicich et al., 2013; Gačić et

al., 2014; Querin et al., 2016). This discharge event, which took place in 2012, generated an abrupt and temporary inversion of the upper-layer Ionian circulation from cyclonic to anticyclonic circulation. Gačić et al. (2014) were able to accurately determine the initiation and cessation of dense water flow due to the ample availability of in situ data mainly from floats. They estimated that the sudden inversion of the horizontal circulation from cyclonic to anticyclonic took place in June 2012 and the subsequent return to cyclonic conditions occurred in February/March 2013. To carry out the comparison with the tank experiment, we determine the

response of the surface geostrophic flow field to the dense water discharge event from satellite altimetry data. In addition, we use the density field during the event, as well as the subsurface flow, from the CMEMS Reanalysis. We simulate the real situation by releasing dense water in the tank for a limited time interval during the experiment (45 rotation days). As mentioned earlier, the flow field response in the tank was observed from the current data, while the density field variations were detected from the vertical profiling in the central deep area of the tank.

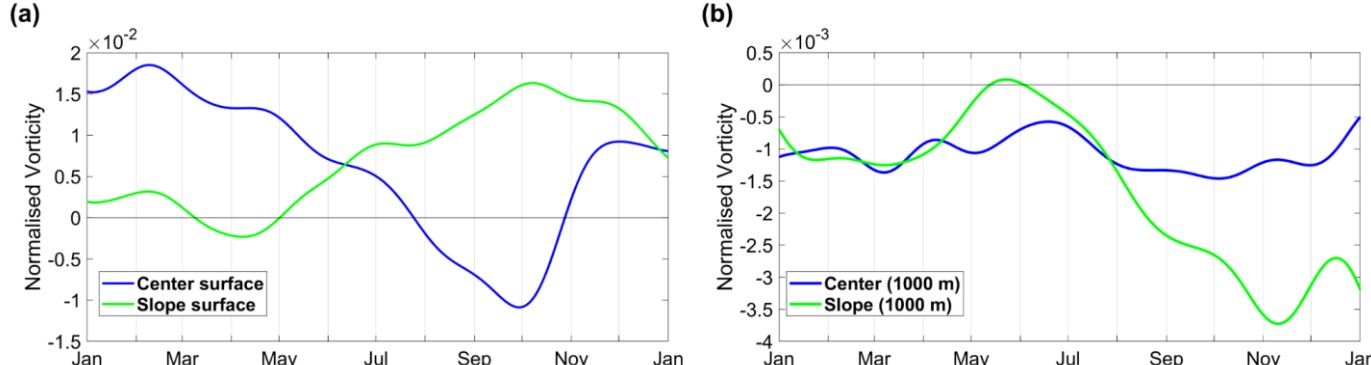

**Figure 11: The northern Ionian Sea in 2012: temporal evolution of the vorticity normalized by the Coriolis parameter $f$ ($10^{-4}$ s$^{-1}$) in the surface (a) and deep (b) layers (1000 m depth) of the slope and in the center. Please note the different scales of the surface (a) and deep (b) vorticity values.**

        We define the continental slope in the northern Ionian Sea as the area between the 2200 m isobath and the coast, while the part equivalent to the flat-bottom central deep area of the tank, is represented by the rest of the basin within latitudinal limits of 37-40°N (Fig. 1a). Indeed, the vorticity evolution in the northern Ionian Sea in 2012 (Fig. 11) shows behavior similar to that in the tank (see Fig. 5, experiment 27). Concomitantly, with the start of the dense water flow in approximately May 2012, the vorticity
in the surface layer of the slope area inverted (Fig. 11a) and remained cyclonic for the rest of the year. At the same time, in the surface layer of the central area, weakening of the cyclonic vorticity took place, which eventually resulted in prevalent anticyclonic motion in approximately July. The flow fields in the surface layer of both the continental slope and central area reached their maximum cyclonic and anticyclonic motion, respectively, in late September to early October (Fig. 11a) when the dense water outflow from the Adriatic seemingly ceased, as suggested by Gačić et al. (2014). At 1000 m depth in the continental slope area,
the vorticity curve (Fig. 11b) varied out of phase with respect to the upper layer (Fig. 11a). This suggests the presence of a two-layer structure in the vorticity field in the continental slope area, similar to what was observed in the rotating tank experiments. In the central area, as in the rotating tank experiments, no clear two-layer structure occurred in the vorticity field and only in the surface layer the circulation inverted from cyclonic to anticyclonic motion (Fig. 11a). The 1000 m vorticity values are smaller than those at the surface since the depth of 1000 m is below the velocity zero-crossing level but close to it. According to some estimates
from the thermal wind relationship (Giuseppe Civitarese, personal communication), the zero-crossing is situated at a depth of approximately 800 m. Consequently, velocities are smaller than those at the surface, resulting in smaller vorticity values, even by one order of magnitude.

By comparing the outputs of the CMEMS Reanalysis (which assimilates the in situ data) and the vertical profiles of the ARGO floats that were active in the northern Ionian during 2012, we observe that the reanalysis absolute density values are typically larger than float densities (not shown). However, the temporal variations in both data sets are consistent. Thus, we reconstruct the evolution of the density fields on the continental slope and in the deep area of the northern Ionian Sea using only the data from the CMEMS Reanalysis and compare it with the vorticity variations at the surface and in the deep layer (i.e., 1000 m depth, Fig. 12) over a slightly longer time interval than in Fig. 11. From the evolution of the vorticity field in the surface layer and the vertical density distribution in subareas A and B, it is evident that in the open sea, the cyclonic vorticity started decreasing concurrently with the rise of the isopycnals (below 800 m depth) and squeezing of the upper layer, eventually becoming anticyclonic in approximately August 2012 (Fig. 12b). Conversely, the upper-layer isopycnals (e.g., 29.2 kg m$^{-3}$) on the continental slope area deepened, and the anticyclonic vorticity started to decrease, eventually switching to cyclonic circulation due to the dense water sinking and surface layer stretching (Fig. 12a), as mentioned above for the rotating tank experiments.

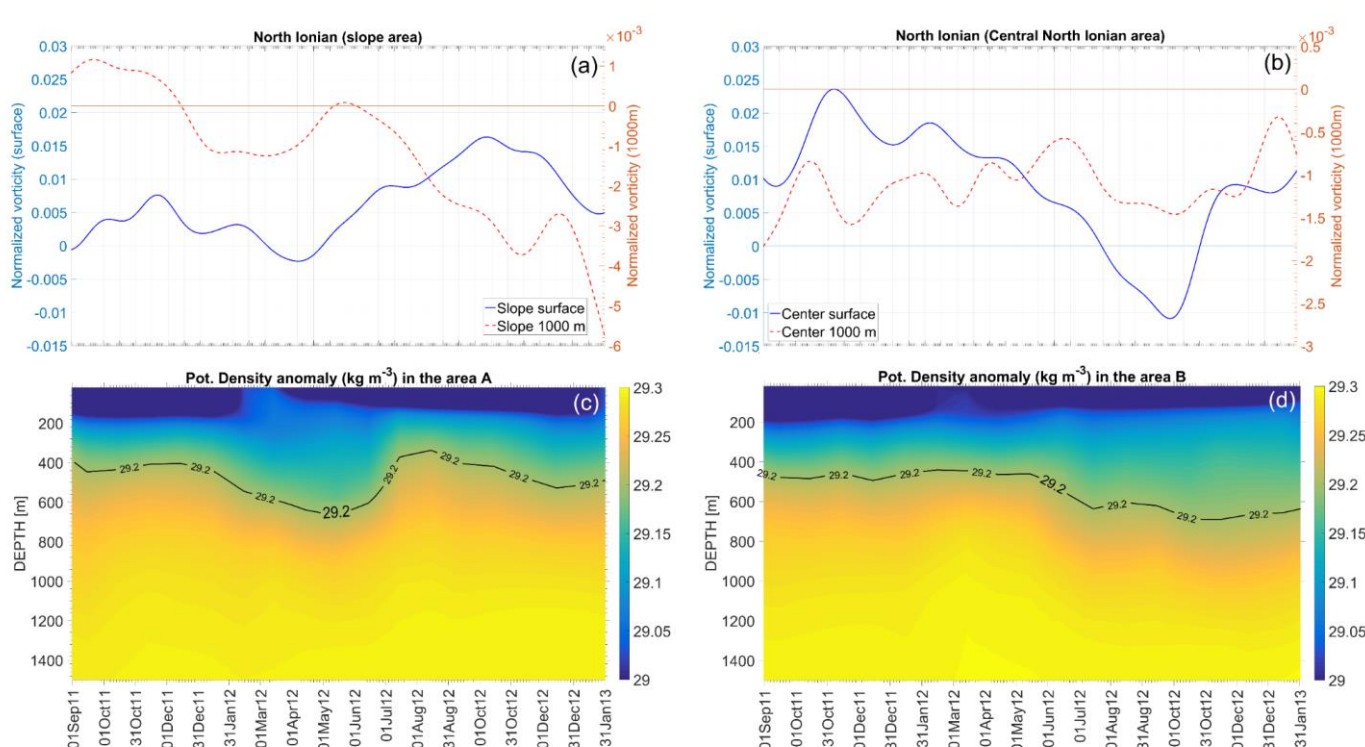

**Figure 12: The Ionian Sea, continental slope - area A and center (deep open sea) - area B (see Fig. 1): temporal evolution of the vorticity normalized by the Coriolis parameter $f$ (10$^{-4}$ s$^{-1}$) at surface and 1000 m depth over the slope (a) and in the center (b); the Hovmöller diagram of the potential density anomaly over the slope (c) and in the center (d).**

The dynamic similarity between the Ionian Sea and the laboratory experiments was discussed in Rubino et al. (2020) by comparing the Burger number in the laboratory and oceanic conditions. Also, as evidenced in the same paper, another important

similarity that should exist is between the in situ and laboratory ratios of the topographic slope and the initial geostrophic slope, which means that the non-dimensional number $g's(fV)^{-1}$, must be preserved in the laboratory.

The similarity between the Ionian Sea and the rotating tank can also be quantified by comparing the respective vorticity rate of change, i.e., $\frac{\partial \zeta}{\partial t}$. It is inversely proportional to the residence time of the upper layer (see Eq. 5). Hence, the ratio of the inclination of the vorticity curve for the Ionian Sea and the rotating tank is inversely proportional to the ratio of their residence times associated with the dense water flow rate. Calculating the receiving volume of the Ionian Sea and the tank and knowing the dense water flow rate in the tank in experiment 27 ($1.6\ 10^{-3}$ m$^3$ s$^{-1}$) and the dense water outflow from the Adriatic (on average 3 $10^5$ m$^3$ s$^{-1}$ according to Lascaratos, 1993), we estimate the ratio between the residence times. On the other hand, we estimate $\frac{\partial \zeta}{\partial t}$ from vorticity curves both for the rotating tank and the Ionian (see Figs. 5 and 11) and the ratio of the vorticity rate of change of the two basins. Our results indeed show that the ratio between residence times of the Ionian Sea and the rotating tank is of the same order of magnitude as the ratio of the vorticity rate of change in the rotating tank and in the Ionian Sea. This confirms the dynamic similarities of the dense water flow in the two basins.

## 5. Summary and conclusions

The decadal inversions of the horizontal circulation, peculiar phenomena in the Ionian Sea, according to the BiOS theory (see, e.g., Rubino et al., 2020 and papers cited therein), are not wind induced but are due to inversions of the internal density gradients. Observations reveal that a reversal can occur very rapidly, i.e., at time scales on the order of a month (see Gačić et al., 2014). Here, we simulate this type of situation in a rotating tank and compare it with observational data gathered in the Ionian basin during the 2012 exceptional dense water overflow. This remarkable phenomenon occurred when, due to the harsh 2012 winter, the BiOS cyclonic mode, which started in 2011, was suddenly interrupted and reversed to anticyclonic flow. To carry out the comparison between such reversal and tank experiments, we focus on two laboratory experiments where different dense water discharge rates created dynamics similar to those observed in the real ocean. For the two experiments analyzed in detail, the ambient fluid consists of two layers: the upper layer is made of freshwater, while the lower layer has a density of 1015 kg m$^{-3}$. In the first part of the experiments, 1010 kg m$^{-3}$ water was discharged for a period of 45 rotation days (1 day = 120 s), after which water with a high density of 1020 kg m$^{-3}$ was released until the 90$^{th}$ day. We varied the dense water flow rates of the two experiments and observed the evolution of the current field. The formation of the large basin-wide anticyclonic gyre in the surface layer of the central flat-bottom area of the tank initiates after the dense water flow starts. Concurrently, over the slope area in the upper layer, the cyclonic vorticity manifests as a series of counterclockwise-traveling mesoscale cyclones (leaving the shallow water on their right) or in the form of cyclonic basin-wide shear. We show that mesoscale eddy activity depends on the dense water discharge rate. Additionally, mesoscale eddies propagate anticlockwise from dense water sources until the onset of basin-wide anticyclonic circulation. Then, the vortices are advected by the mean basin-wide flow in the opposite direction. In the lower layer of the slope area, instead, an anticyclonic vorticity is generated, and therefore, in that portion of the tank, the current field behaves in a two-layer fashion from the point of view of the vorticity pattern. The vorticity in the Ionian Sea shows a vertical structure both in the continental slope and in the central deep area, as in the rotating tank. We show that the evolution of the flow field in the Ionian

following dense water outflow from the Adriatic is dynamically similar to the flow field in the rotating tank following dense water injection. A similarity is shown for the experiment with a dense water discharge rate of $1.6 \ 10^{-3}$ m$^3$ s$^{-1}$ when the ratio between the vorticity rate of change in the Ionian and tank is of the same order of magnitude as the inverse of the ratio of the residence times. This laboratory experiment confirms that the internal forcing, the only forcing applied in the rotating tank, is sufficient to create inversions of the basin-wide cyclonic circulation to anticyclonic circulation in the Ionian Sea, as already hypothesized by the BiOS

theory.

**Appendix A:**

Turbulent diffusion could be estimated both in the central deep and the slope domain. In both areas the turbulent diffusivity can be associated with a stratified shear layer. The turbulent energy of a typical eddy of size $L$ in this shear layer is of the order $E_t \sim (\varepsilon L)^{2/3}$, with $\varepsilon$ being the dissipation rate in m$^2$ s$^{-1}$, whereas the energy associated with buoyancy and shear is $E_b \sim (NL)^2$ and $E_s \sim (L \ \partial u/\partial z)^2$, respectively. Here $N^2 = -g\rho^{-1} \ (\partial \rho/\partial z)$ is the Brunt-Väisälä frequency and $\partial u/\partial z$ is the vertical shear.

Balancing turbulent and forcing components yields a buoyancy length (i.e., the Ozmidov length scale) $L_o = (\varepsilon N^{-3})^{1/2}$ and a

560 shear length $L_s = (\varepsilon <\partial u/\partial z>^{-3})^{1/2}$. This latter scale was defined initially by Corrsin (1958) as the smallest scale at which anisotropy effects resulting from a large-scale shear are carried out by the turbulent cascade.

For low Richardson numbers the effect of shear dominates the effect of buoyancy therefore, the relevant quantity to define the mixing scale is shear, while for large Richardson numbers the relevant quantity is the Brunt-Väisälä frequency N. The smaller of these lengths limits the typical eddy size.

Odier et al. (2012) proposed an approach based on the Prandtl mixing length model with a characteristic mixing length $L_m$ to relate the turbulent eddy diffusivity $v_t$ to the velocity fluctuations and gradients in a stratified shear layer. They showed that $L_m$ was proportional to $L_s$ so that the turbulent diffusivity $v_t$ can be evaluated using:

$$v_t \sim L_s^2 <\frac{\partial u}{\partial z}>.$$

In the tank experiments we estimate $\varepsilon \approx O (10^{-4}$ m$^2$ s$^{-3})$ from the PIV measurements, $N=0.1$ s$^{-1}$, and $<\partial u/\partial z> \approx O(1 \ \text{s}^{-1})$. Introducing these values in the above expressions we obtain for $L_o \approx 0.3$ m and for $L_s \approx 0.01$ m. Hence, the eddy length scale is determined by shear since $L_s < L_o$ , so that our estimate gives $v_t \sim 10^{-4}$ m$^2$ s$^{-1}$ for the tank experiments. Using a velocity scale, the Nof speed $U = 0.1$ m s$^{-1}$ and the vertical scale of the gravity current of $h = 0.05$ m, the normalized turbulent

diffusivity is $v_t(Uh)^{-1} \sim 0.02$. We also expect that at higher Richardson numbers, as typical in the ocean, the length scale will be determined by buoyancy, since $L_o < L_s$ so our oceanic estimate may be slightly higher. Parameterizations in ocean models (Lane-

Serff and Baines, 2000) have used values in the range $0.032 < v_t < 0.70$ m$^2$ s$^{-1}$ for typical overflow scenarios. Critical to extrapolating to oceanic conditions is a systematic exploration of the dependence of the mixing lengths on turbulence intensity and on the degree of stratification as measured by the Richardson number.

Lane-Serff and Baines (2000) also proposed a relation to evaluate turbulent diffusivity based on scales that are easier to extrapolate for oceanic overflows, which reads:

$$v_t = (4k^2 Q f^2)(g's^2)^{-1},$$

where $Q$ is the injected volume transport of the gravity current and $k$ is taken to be 2.5 x 10$^{-3}$ (a typical value for oceanographic flows, e.g., Lane-Serff 1993, 1995; Bombosch and Jenkins 1995). The value of the Adriatic outflow into the Ionian Sea $Q \approx 10^5$ m$^3$ s$^{-1}$, along with $f = 10^{-4}$ s$^{-1}$, $g' = 0.003$ m s$^{-2}$ and $s = 0.02$ give $v_t \approx 0.02$ m$^2$ s$^{-1}$. Rare in situ observations of upper ocean turbulent mixing, stratification and currents in the Adriatic Sea resulted in the estimates of the eddy diffusivity in the central part of the basin (Peters and Orlić, 2005). The values are however smaller than those presented above due to weak shear and strong

stratification combined with large Richardson numbers. Using $U = 0.1$ m s$^{-1}$ as a typical velocity scale and a typical vertical length scale of the overflow of $h = 200$ m, one obtains a normalized turbulent diffusivity of $v_t (Uh)^{-1} \approx 0.001$ for the real flow conditions in the Ionian Sea, which is smaller than the laboratory value (0.02). Note that if the above expression of Lane-Serff and Baines (2000) for the oceanic environment is applied to the laboratory experiments, the value of the constant $k$ should be adapted to larger values, since the Reynolds number is smaller in the experimental conditions than in the real ocean (see Lane-Serff 1993, 1995;

Bombosch and Jenkins 1995).

**Appendix B:**

        The theoretical curve in Fig. 4 obtained from Eq. 3 on the lower layer thickness $h$ fits rather well the experimental data

from the single-point density measurements at Cp3 site. Few departures from the theoretical curve are however evident, like e.g. the anomaly seen in the MLD (Fig. 3, reported also in Fig. B1a), in the rate of change of the lower layer thickness (Fig. 4), and in the vorticity evolution (Fig. 6) for EXP 27, around rotation day 75$^{th}$. Concurrently, at the Cp3 site, the local maximums are present both in the radial and tangential velocities in the deep layer (Figs. B1 b and c). These features are linked to the passage of a mesoscale anticyclonic eddy. Indeed, the formation and passage of the eddy in the vicinity of Cp3 is clearly evident from the

horizontal distributions of the velocity vectors in the lower layer beneath the pycnocline, represented by the level 11 (Fig. B2). Overall effects of the eddy passage are: an anomaly in the vertical density distribution and the transient thickening of the pycnocline layer, an anomaly in the relationship between the vorticity rate of change and the rate of change of the bottom layer thickness (Figs. 3 and 4). This event is however of limited spatial and temporal extent. In fact, it should be kept in mind that such an anomaly has been detected because the mesoscale eddy was passing through the Cp3 site. Moreover, less prominent features

are observed at other rotation days, probably generated by similar mesoscale features passing close but not over the Cp3 location. It is plausible that some similar eddies have not been detected at all, because travelling out of range of Cp3 site, during EXP 27, but also during other experiments.

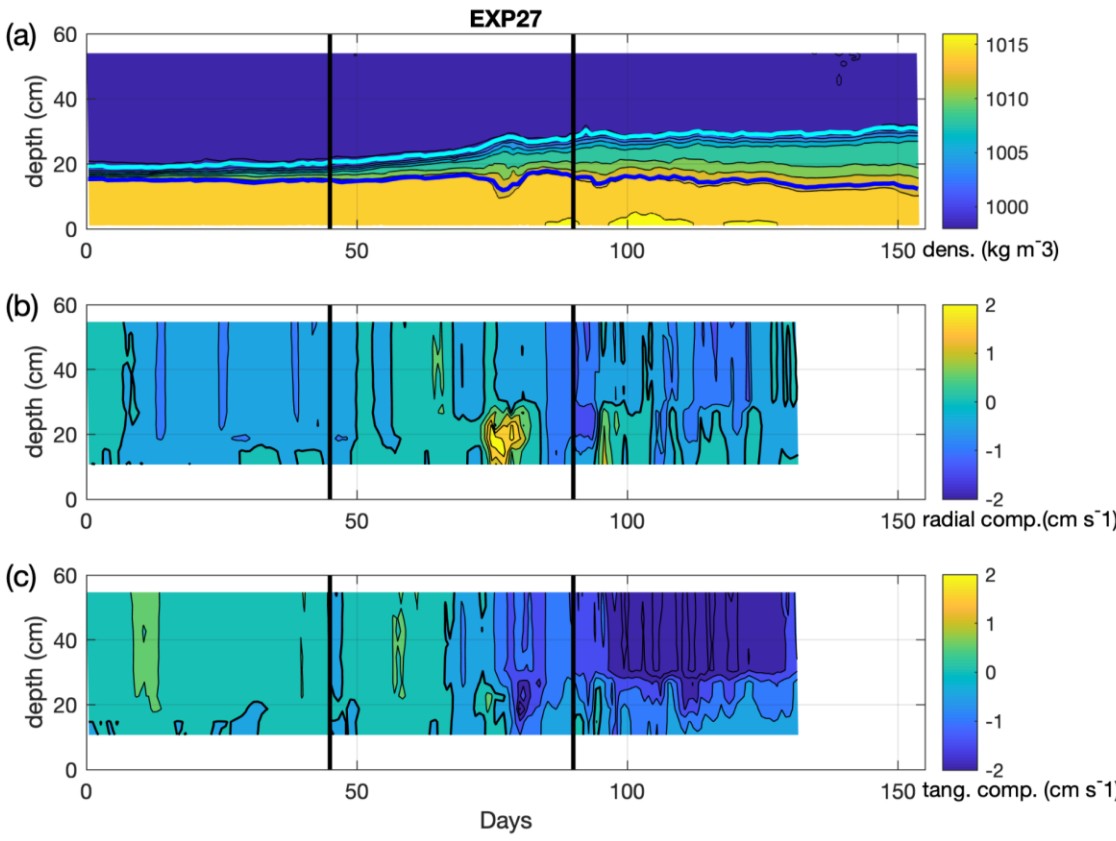

**Figure B1: Hovmöller diagram of density (a), radial (b) and tangential (c) velocity components for EXP 27. The black isoline interval in panel (a) is 2 kg m⁻³, starting from 1016 kg m⁻³ at the bottom; thick cyan and blue lines denote MLD and the base of the pycnocline, respectively. Bold isolines in (b) and (c) indicate 0 cm s⁻¹, and the isoline interval is 0.5 cm s⁻¹. Black solid vertical lines indicate rotation**
**days 45 and 90, i.e., the start and end of the high-density water injection, respectively (for reference see Fig. 2 and Tables 1 and 2).**

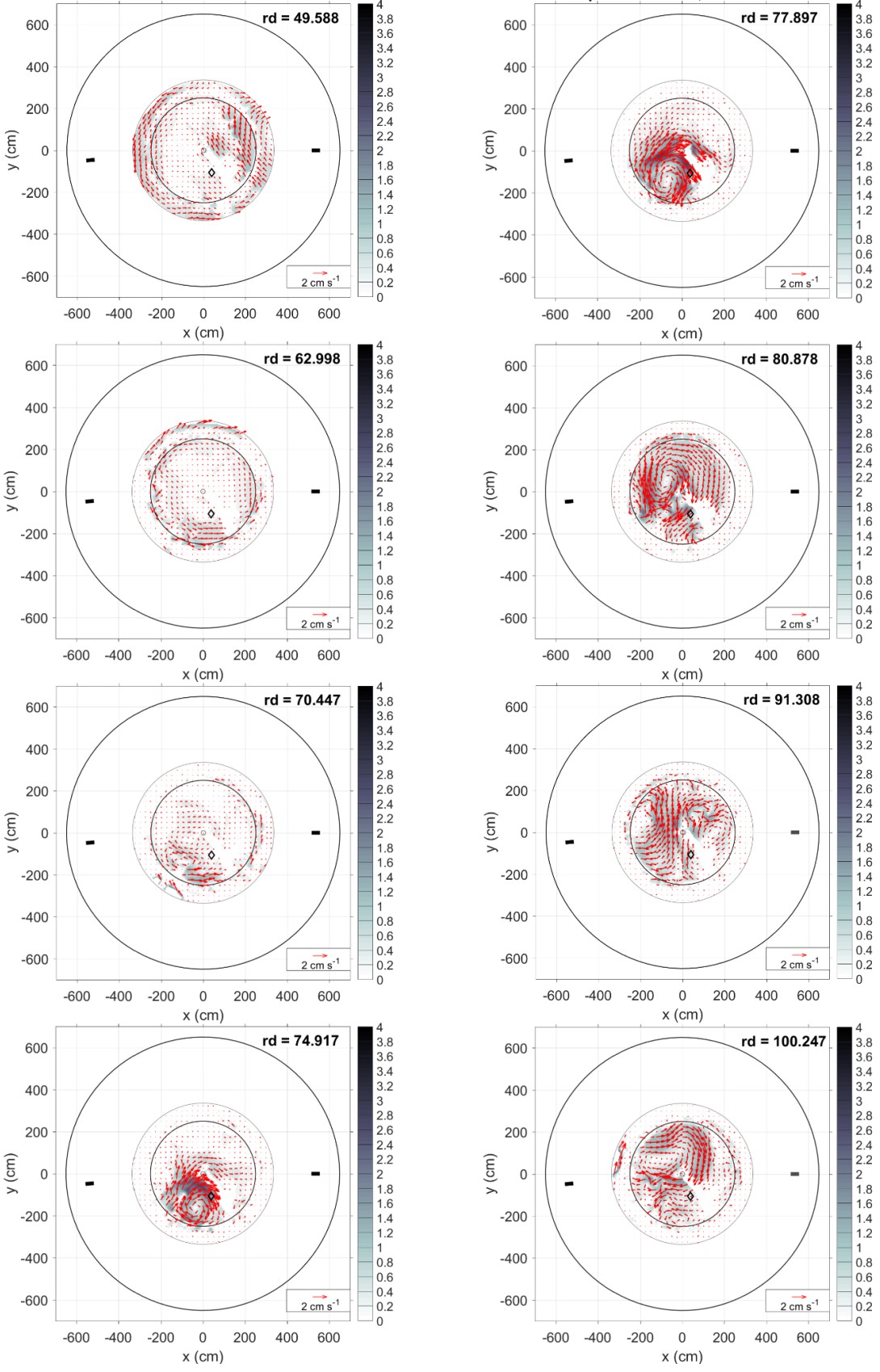

**Figure B2: Evolution of the flow field in the lower layer of the rotating tank (represented by level 11) during experiment 27. The rotation days are indicated inside each panel. The diamond shows the location of the conductivity probe Cp3, and the two bars show the location of the two dense water sources. The outer and inner bold circles delimit the tank edge and the central deep area, respectively. Grey circle delimits the extension of the level 11.**

### Data availability

All data sets used can be made available by request to the first and corresponding author.

### Supplement

Link to S1&S2.zip

### Authors contribution

MG, AR, and JS designed the laboratory experiments; MG prepared the manuscript with the help of all coauthors; AR, LU, VK, VM, MEN, VC, MO, JS, and MM contributed to writing and editing of the manuscript and participated in the theoretical aspect discussions; LU, VK, MM, MB, MEN, VC, and RVB carried out the data analysis using specifically designed analysis routines; and VK, MB, MEN, VC, JS, RVB, SV, BP, GS, MG, and AR performed the laboratory experiments.

### Competing interest

The authors declare that they have no known competing financial interests or personal relationships that could have appeared to influence the work reported in this paper.

### Special issue statement

### Acknowledgments

We greatly appreciate the technical and scientific support offered by the LEGI staff during the project implementation. We thank G. Civitarese for his enthusiasm and important contribution to the early work in the project preparation. We are grateful to A. Wirth for making available his design of the injectors used in the experiments and for the useful discussions on the vorticity dynamics. Finally, our thanks go to P. Del Negro for her encouragement and interest in our project. Last but not the least, the anonymous reviewers contributed greatly to the quality of the paper. We also thank the anonymous reviewers for their careful reading of our manuscript and their many insightful comments and suggestions.

**Financial support**

The project BiOS - CRoPEx has received funding from the European Union's Horizon 2020 research and innovation program under grant agreement No. 654110, HYDRALAB+.

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
