# Peer review of "Impact of dense water flow over a sloping bottom on open-sea circulation: Laboratory experiments and the Ionian Sea (Mediterranean) example"

_Ocean Science, 2020_

## Referee Comment (RC3)

Review of the paper entitle:

"Impact of the dense water flow over the sloping bottom on the open-sea circulation: Laboratory experiments and the Ionian Sea (Mediterranean) example"
by
Miroslav Gačić, Laura Ursella, Vedrana Kovačević, Milena Menna, Vlado Malačič, Manuel Bensi, Maria-Eletta Negretti, Vanessa Cardin, Mirko Orlić, Joël Sommeria, Ricardo Viana Barreto, Samuel Viboud, Thomas Valran, Boris Petelin, Giuseppe Siena, Angelo Rubino.

General comments.

The scientific matter of the manuscript isn't a really new; actually in the literature there are many example on this, either in the modelling field or in the analysis of the in situ observations and correctly in the manuscript there are a long list of key references. However, the manuscript tackle an important and new relevant scientific issue dealing with the analysis of the ocean processes related to the propagation, spreading and adjustment of the density anomaly in a complex topography like as it is the Ionian Mediterranean sub-basin.

This study, specifically the tank-experiment, reveals the important role of the n-layer stratification in the vortex rotation within the framework quasi-geostrophic model on an f-plane.

For all these reasons that the results of this paper are very interesting for the oceanographic communities and in particular for those scientists more implicated on the Mediterranean studies.

However the present version of the manuscript have a lot of a weak points and therefore is not ready to be published for the following reason.

Major revision:

Among the weak points the following ones is the most relevant:

•       The manuscript encompassing a comprehensive introduction of the experimental apparatus and methodology followed by the authors and every thing is very well conducted, except the relative role (scale ratio) of the central part of the tank respect to the sloping part. In fact, looking the figure 7 in the manuscript seems that the dynamics driven by the slope domain dominates on those generate in the flat domain, making very difficult distinguishing the difference between the two dynamics. This isn't irrelevant to make more realistic the comparison with the northern Ionian Sea circulation in section 4.
•       Moreover is very confusing the theoretical and modelling equations that are used to analyse the experimental results: the equation 1 is not the same used by the cited paper of Lee-Lueng, F. and Davidson, R. A. (A note on the barotropic response of sea level to time-dependent wind forcing. J. Geophys. Res., 100, C2, 24955-24963, 1995) that use a classical linear barotropic vorticity equation, may be the authors have to use a different reference.
•       However, the most relevant matter is related to the stratification that, at the end, is the core problem of the manuscript. It is well know that a good representation of the ocean dynamics is a three-layer system and this is this is confirmed even in this case as is

well evident in the figure 2c (experiment 27), specifically around the 75th day in which we see the respond of the pressure to the injection of the density anomaly and subsequence stratification in three layer (or a continuously stratification see references), is very interesting the impact of the redistribution of density and pressure within the water column in the figure 3 (and also figure 5) experiment 27 at the same day (around the 75th). These figures are the most interesting of the manuscript and at the same time are those that demonstrate the weakness of the theory presented in this manuscript: actually can't demonstrate the opposite vorticity at the 75th day and the corresponding kinetic energy anomaly in the lower layers. However, despite this experimental evidence and the same recognition as the authors themselves that the dynamics follow at least a two-layers system, even so at the end the equation that the authors used is written in a one-layer formulation. This is not irrelevant for physical point of view. Is matter of fact that dealing with one, two or three layer formulation of the QG equation, produce a different vorticity relation between the several interfaces along the water column (see Sokolovskiy paper and all reference herein). This is true either in the flat or in the slope domain and finally on the comparison with the realistic example of the Ionian sea.

In conclusion the circular rotating tank experiment shows in an impressing way (this could be more impressing with a different scale ratio between the slope/flat domain), the adjustment of the vorticity along the continuously stratified water column (and its dependence from the layers-thickness distribution) when it is subjected to a density anomalies: first along the slope and afterwards during the spreading of the anomaly density flow along the flat bottom; finally is very arduous to do some comparison, in the present version of the manuscript, between the tank experiment and what was observed in the northern Ionian sea in 2012.

Minor revision:

- Line 326 "level 1" is referred to inclined laser sheet levels 1 of the Figure 1?
- Line 451 at which model is referred? Please give more details;
- figure 1 the word "cp3" is not clear;
- figure 2 in the color tab 0-15 means that the range of density is between 1000-1015?

References.

1. M. A. Sokolovskiy; Stability of an Axisymmetric Three-Layer Vortex, Izvestiya, Atmospheric and Oceanic Physics, Vol. 33, No. 1, 1997, pp. 16–26. Translated from Izvestiya AN. Fizika Atmosfery i Okeana, Vol. 33, No. 1, 1997, pp. 19–30. (and all references herein)

---

## Author Comment (AC1)

**Response to the comments of the Ref. 3**

**C.** 1: The manuscript encompassing a comprehensive introduction of the experimental apparatus and methodology followed by the authors and everything is very well conducted, except the relative role (scale ratio) of the central part of the tank respect to the sloping part. In fact, looking the figure 7 in the manuscript seems that the dynamics driven by the slope domain dominates on those generate in the flat domain, making difficult distinguishing the difference between the two dynamics. This isn't irrelevant to make more realistic the comparison with the northern Ionian Sea circulation in section 4.

Reply: We thank the Referee very much for this comment and as our reply we added Appendix A (text below) where we present the similarity criteria between the laboratory experiment and the real ocean.

**"Appendix A: Dynamical similarity**

The fundamental part of the experimental design is to achieve a "dynamical similarity" between the real-ocean and laboratory. In our case, to simulate the Adriatic overflow into the Ionian basin and to reproduce the North Ionian Gyre (NIG) reversals in the laboratory, two relevant non-dimensional numbers are considered to evaluate the dynamical similarity.

First, the Burger number gives the ratio between the internal Rossby radius of deformation and the geometrical scale of the Ionian basin/the rotating table for the experiment. Considering the depths of both, the laboratory and Ionian basin, the Coriolis parameter f and the density anomaly, expressed using the reduced gravity g', we obtain a similarity between ocean and laboratory phenomena from the experimental values indicated in section 2.1 Experimental design of the manuscript and in the Appendix B (see below). In particular, the combination of those values with an experimental slope of s=0.1 yields a Burger number of  $B_u=0.1$  like the one observed in the ocean.

Another important similarity that should exist is between the in situ and laboratory ratios of the topographic slope and the initial geostrophic slope, which means that the non-dimensional number  $S=g's(fV)^{-1}$ , with V the initial (Adriatic) overflow velocity, must be preserved in the laboratory experiments. Considering these values for both the Ionian basin and Adriatic outflow and the similarity of Burger number, we selected the topographic slope of 0.1 in order to fall within the similarity values of the in situ conditions ranging approximately as 2.4 < S < 9.4.

*Finally, the experiments also preserved dynamical similarity accounting for frictional effects by considering the Ekman non-dimensional numbers.*"

**"Appendix B: Turbulent diffusion**

The turbulent diffusion could be estimated both in the central deep and the slope domain. In both areas the turbulent diffusivity can be associated with a stratified shear layer. The energy of a typical eddy of size L in this shear layer is of the order  $E_t \sim (\varepsilon L)^{2/3}$ , with  $\varepsilon$  being the dissipation rate in  $m^2s^{-3}$ , whereas the energy associated with buoyancy and shear is  $E_b \sim (NL)^2$  and  $E_s \sim (\partial u/\partial zL)^2$ , respectively. Here  $N^2 = -g(\partial \rho/\partial z)/\rho$  is the Brunt-Väisälä frequency and  $\partial u/\partial z$  is the vertical shear.

Balancing turbulent and forcing components yields a buoyancy length (i.e., the Ozmidov length scale)  $L_o = (\varepsilon N^{-3})^{1/2}$  and a shear length  $L_s = (\varepsilon < \partial u/\partial z >^{-3})^{1/2}$ . This latter scale was defined

initially by Corrsin (1958) as the smallest scale at which anisotropy effects resulting from a large-scale shear are carried out by the turbulent cascade.

For low Richardson numbers the effect of shear dominates the effect of buoyancy, therefore the relevant quantity to define the mixing scale is the shear, while for large Richardson numbers the relevant quantity is the Brunt-Väisälä frequency N. The smaller of these lengths limits the typical eddy size.

Odier et al. (2012) proposed a model based on the Prandtl mixing length model with characteristic mixing length  $L_m$  to relate the turbulent eddy diffusivity  $v_t$  to the velocity fluctuations and gradients in a stratified shear layer. They showed that  $L_m$  was proportional to  $L_s$  so that the turbulent diffusivity can be evaluated using:

 $v_t \sim L_s^2 < \partial u / \partial z >$ .

In the tank experiments, we estimate,  $\varepsilon \approx O(10^{-2} \text{ m}^2\text{s}^{-3})$  from the PIV measurements, N=0.1  $\text{s}^{-1}$ ,  $\langle \partial u / \partial z \rangle \approx O(1 \text{ s}^{-1})$ . Introducing these values in the above expression we obtain for  $L_o \approx 3$  m and for  $L_s \approx 0.1$  m; hence, the length scale will be determined by shear since  $L_o > L_s$  so that our estimate gives  $v_t \sim 10^{-4} \text{ m}^2\text{s}^{-1}$  for the tank experiments. Using a velocity scale, the Nof speed U=0.1 ms-1 and the vertical scale of the gravity current of h=0.05 m, the normalized turbulent diffusivity gives  $v_t(Uh)^{-1} \sim 0.02$ . We also expect that at higher Richardson numbers, as typical in the ocean, the length scale will be determined by buoyancy, since  $L_o < L_s$ , so that our oceanic estimate may be a bit high. Parametrizations in ocean models (Lane-Serff and Baines 2000) have used values in the range  $0.032 < v_t < 0.70 \text{ m}^2\text{s}^{-1}$  for typical overflow scenarios. Critical to extrapolating to oceanic conditions is a systematic exploration of the dependence of the mixing lengths on turbulence intensity and on the degree of stratification as measured by the Richardson number.

Lane-Serff and Baines (2000) also proposed a relation to evaluate the turbulent diffusivity based on scales that are easier to extrapolate for oceanic overflows, which reads:

 $v_t = (4k^2 Q f^2) (g' s^2)^{-1},$

where Q is the injected volume transport of the gravity current, k is taken to be 2.5 x  $10^{-3}$  (a typical value for oceanographic flows, e.g., Lane-Serff 1993, 1995; Bombosch and Jenkins 1995). This gives for the values of the Adriatic outflow into the Ionian Sea  $Q\approx 10^{5}$  m3s-1,  $f=10^{-4}$  s-1, g'=0.003 ms-2 and s=0.02 giving  $v_{l}\approx 0.02$  m2s-1. Rare in situ observations of upper ocean turbulent mixing, stratification and currents in the Adriatic Sea resulted in the estimates of the eddy diffusivity in the central part of the basin (Peters and Orlić, 2005). The values are however much smaller than those presented above due to weak shear and strong stratification combined with large Richardson numbers. Using U=0.1 ms-1 as a typical velocity scale and a typical vertical length scale of the overflow of h=200 m, one obtains a normalized turbulent diffusivity of  $v_t(Uh)^{-1} \approx 0.001$  for the real flow conditions in the Ionian Sea, which is smaller than the laboratory value (0.02). Note that if the expression of Lane-Serff and Baines (2000) for the rotating platform conditions is applied to oceanic environment, the value of the constant k needs to be adapted to larger values, since the Reynolds number is smaller in the laboratory conditions than in the real ocean (see Lane-Serff 1993, 1995; Bombosch and Jenkins 1995).

1) Bombosch, A. and Jenkins A. : Modeling the formation and deposition of frazil ice beneath the Filchner–Ronne Ice Shelf. J. Geophys. Res., 100, 6983–6992, 1995.

2) Corrsin S.: Local isotropy in turbulent shear flow, National Advisory Committee for Aeronautics RM 58B11, 1958.

3) Lane-Serff, G. F. : On drag-limited gravity currents., Deep-Sea Res. I, 40, 1699–1702, 1993.

4) Lane-Serff, G. F.: On meltwater under ice-shelves, J. Geophys. Res., 100, 6961–6965, 1995.

5) Lane-Serff, G. F. and P. G. Baines: Eddy formation by overflows in stratified water. J. Phys. Ocean., 30, 327–337, 2000.

6) Odier, P., Chen J. and R. E. Ecke: Understanding and modeling turbulent fluxes and entrainment in a gravity current, Phys. D: Nonlin. Phen., 10.1016/j.physd.2011.07.010, 2012.
7) Peters, H. and M. Orlić: Turbulent mixing in the springtime central Adriatic Sea. Geofizika, 22, 1-19, 2005."

C2 Moreover is very confusing the theoretical and modelling equations that are used to analyze the experimental results: the equation 1 is not the same used by the cited paper of Lee-Lueng, F. and Davidson, R. A. (A note on the barotropic response of sea level to time-dependent wind forcing. J. Geophys. Res., 100, C2, 24955-24963, 1995) that use a classical linear barotropic vorticity equation, may be the authors have to use a different reference.

**Reply: As our response to this comment, we simply removed the reference saying that we will be treating the well-known linear barotropic vorticity equation for an f-plane approximation with the sloped bottom in radial coordinates for the surface layer without wind-stress forcing. We also defined the radial coordinate being perpendicular to isobaths and negative downslope.**

C3 However, the most relevant matter is related to the stratification that, at the end, is the core problem of the manuscript. It is well know that a good representation of the ocean dynamics is a three-layer system and this is this is confirmed even in this case as is well evident in the figure 2c (experiment 27), specifically around the 75th day in which we see the respond of the pressure to the injection of the density anomaly and subsequence stratification in three layer (or a continuously stratification see references), is very interesting the impact of the redistribution of density and pressure within the water column in the figure 3 (and also figure 5) experiment 27 at the same day (around the 75th). These figures are the most interesting of the manuscript and at the same time are those that demonstrate the weakness of the theory presented in this manuscript: actually, can't demonstrate the opposite vorticity at the 75th day and the corresponding kinetic energy anomaly in the lower layers. However, despite this experimental evidence and the same recognition as the authors themselves that the dynamics follow at least a two-layers system, even so at the end the equation that the authors used is written in a one-layer formulation. This is not irrelevant for physical point of view. Is matter of fact that dealing with one-, two- or three-layer formulation of the QG equation, produce a different vorticity relation between the several interfaces along the water column (see Sokolovskiy paper and all reference herein). This is true either in the flat or in the slope domain and finally on the comparison with the realistic example of the Ionian Sea. In conclusion, the circular rotating tank experiment shows in an impressing way (this could be more impressing with a different scale ratio between the slope/flat domain), the adjustment of the vorticity along the continuously stratified water column (and its dependence from the layers-thickness distribution) when it is subjected to a density anomalies: first along the

slope and afterwards during the spreading of the anomaly density flow along the flat bottom; finally is very arduous to do some comparison, in the present version of the manuscript, between the tank experiment and what was observed in the northern Ionian sea in 2012.

Reply: The Reviewer is right when he argues that the evolution of a 2-layer and 3-layer system differs. However, in the present paper we do not attempt to determine the evolution of the eddying dynamics of the system. Our focus is on the dynamics of two homogeneous layers and the relationship between their thickness and the relative vorticity which is constrained for every layer by the conservation of potential vorticity (PV) in each of them, independently of the dynamics above and below it. So, the conservation of PV is ensured in a single layer, and it does not depend on how many layers stay above or below. In addition, we reduce the effect of the horizontal advection on our analysis by considering horizontal averages over larger areas (central and slope). Please, note also that the flow is not advected from one layer to the other.

We thank the Reviewer for highlighting the event of the 75th day in experiment 27 which is really a special event. We examine more carefully the flow pattern evolution during that event (please, see Fig. R1). One can see the isolated maximums of the radial and tangential velocities in the deep layer at the site of the density profiling (Cp3). These are linked to the passage of a mesoscale anticyclonic eddy around the 75th rotation day in the vicinity of the vertical profiling sensor. Indeed, the formation and passage of the eddy in the vicinity of the vertical profiling site is clearly seen from the horizontal distributions of the velocity vectors (Fig. R2). This eddy then causes an anomaly in the vertical density distribution and the transient thickening of the pycnocline layer, as well as in the relationship between the vorticity rate of change and the rate of change of the bottom layer thickness (see Fig. 3 in the original paper text). For the rest of the experiment the PV equation for the flat bottom describes adequately the dynamics of the bottom layer.

Fig R1: Hovmoller diagram of the density evolution (upper panel), of the radial (middle panel) and the tangential velocity component (bottom panel) for experiment 27.

---

## Author Comment (AC2)

**Response to the comments of the Ref. 1**

*Laboratory experiments are used to investigate the effects of injecting fluid at various flow rates and densities on a sloping boundary into a two-layer stratification in a rotating system. The aim is to examine the effect of overflow waters into the Ionian Sea, and in particular how changes in the density or flow rate can affect the overall circulation within the basin. This is a substantial experimental effort, conducted carefully, and the work is mostly described well, with an analysis of the partitioning between eddy and mean flow KE, the resulting flow within the basin, and comparisons with numerical model output for the Ionian Sea, assimilating in-situ data for 2012. Overall this is a valuable piece of work. There are some aspects of the presentation that should be improved before the paper is published.*

*1 While I realise details are included in other references, you need to give some more details about how the velocities are calculated from the laboratory experiments. An image showing a velocity field earlier in the paper, to help explain the main features of the laboratory flow, would also be useful.*

*In section "2. Data and methods" under the paragraph "2.1 Experimental design", we added the two following sentences describing the methods employed to calculate the velocity.*

*"Sequences of the images at each of the 12 levels were taken with a high-resolution Nikon Camera synchronized with a profiling laser system. It illuminated the Polyamide particles (Orgasol) with a mean diameter of 60 μm and a density of 1.020 kg m$^{-3}$ dispersed in the tank and in the injected saline solutions to allow optical velocity measurements. Velocity fields were computed from the images using a cross-correlation particle image velocimetry (PIV) algorithm encoded with the software UVMAT developed at LEGI. "*

*In addition, the supplementary material S1 shows the main features of the laboratory flow.*

*2 You mention viscous bottom draining (line 313) but there was no mention of this before – something on this should appear in the Introduction.*

*We thank the referee for this observation. In the Introduction, after line 95 we added the following paragraph:*

*"On the slope area, the injected saline solution induces a gravity current whose body quickly reaches an almost geostrophic equilibrium due also to the particular injection method employed. The gravity current consists then of two parts: the first one is the proper 'vein', characterized by an almost along-slope velocity, and the second is a viscous bottom layer, also called Ekman leakage, showing an almost down-slope velocity. A detailed description of the structure of a rotating gravity current composed of a vein and an Ekman leakage is given in Wirth, 2009 (see also Cenedese et al., 2001)."*

*3 It would be useful to have a chart of the Ionian Sea sooner in the paper, and you should mark the locations of the main inflows and sketch the typical circulation(s).*

*We have inserted a new Fig. 1 in the paper showing the map of the Ionian Sea and the sketch of the laboratory tank:*

[Figure]

*Figure 1: (a) Map of the study area in the Ionian basin with a simplified circulation scheme, which changes accordingly to the BiOS regime. Grey horizontal lines indicate the geographical limits within which the mean vorticities above and below the 2200m isobath were calculated. Rectangles A and B indicate the areas where density data (CMEMS reanalysis) were averaged. Concentric rings represent the simplified laboratory tank scheme. Acronyms: AW = Atlantic Water, LIW = Levantine Intermediate Water, AdDW = Adriatic Deep Water; (b) a view of the tank: the slope area is between the red and blue, deep flat-bottom area is inside the blue ring. Dense water injectors are placed at IS1 and IS2. A diamond near the centre shows a location of the Cp3 profiler. Concentric grey rings indicate intersections of the laser sheet levels with the slope. Grey dots indicate a regular x-y grid for tank velocity field (subsampled every 5 nodes for clarity). The map in (a) was created from the bathymetry data ETOPO2v2, NOAA, World Data Service for Geophysics, Boulder, June 2006, doi: 10.7289/V5J1012Q) using the MATLAB software.*

*4 You need to explain how you calculate MKE and EKE in more detail.*

*The explanation about the calculation of MKE and EKE was added in section "2.2 Data Analysis", as follows:*

*"Mean Kinetic Energy (MKE) and Eddy Kinetic Energy (EKE) are computed for the surface layer over the slope based on the time series of current velocity components $v_x$ and $v_y$ according to the system in tank coordinates (Fig. 1b). Specifically, we take $v_x$ and $v_y$ as the respective average from the levels 1, 2, 3, and 4 at each grid point. Hence, MKE=½ $(<v_x>^2 + <v_y>^2)$ and EKE= ½$(<v_x'^2> + <v_y'^2>)$, where the symbol <> means the temporal average in each grid point, $v_x'=v_x$-*

*$<v_x>$, and $v_y'=v_y-<v_y>$. This operation is performed for each measurement phase for which, finally, spatial averages of the MKE and EKE are obtained over the slope area."*

*5 The English needs some attention – below I list a few corrections from the Abstract and Introduction by line number, but there are others, and throughout you often write "the experiment 24" or "the phase II" where "the" should be deleted.*

*30 Density records show*

*56 as happened in*

*69 these studies maintain that*

*80 of dense water in the*

*82 with observations*

*83 with a duration of*

*87 circulation of the open sea*

*91 of vorticity generation*

*Fig 1 (not to scale)*

*Following these comments our paper underwent an English proofread.*

---

## Author Comment (AC3)

**Response to the comments of the Ref. 2**

The manuscript presents a reproduction of an important quasi-decadal mechanism that drives thermohaline oscillations in the Adriatic-Ionian region. I appreciate such a novel approach using tank experiments, which tries to provide the underlying physics that has (still) several possible explanations - this research is important to put proper weight on them (internal forcing vs. wind-stress curl) Therefore, I strongly recommend publication of the manuscript. Still, some issues should be cleared and corrected (also agreeing with comments of Anonymous Referee #1):

- Lines 94-106. It might be more appropriate to have this at the beginning of Section 2 (as an introduction, before Section 2.1), as justifying the applied methodology.

*As suggested by the referee, the text between lines 94 and 106 (see below) was moved after the second paragraph of Section 2.1. :*

*"We distinguish the slope and central deep (flat bottom) areas in the tank that are equivalent to the continental slope and deep zone of the northern Ionian basin, respectively (Fig. 1). We compare the potential vorticity evolution in each area as related to the dense water flow. The two areas are presumably controlled by different processes of the vorticity generation. In the central area (flat bottom) the upper layer squeezing, due to the downslope sinking of the dense water to the lower layer, generates the upper layer anticyclonic vorticity. In the slope area, the upper layer stretching due to the downslope water flow results in the generation of the cyclonic vorticity. The lower layer on the slope is subject to squeezing and anticyclonic vorticity generation as related to the formation of the dense water flow parallel to isobaths."*

- Fig. 1. It might be good to increase the font of the smallest labels, they can be hardly read in such a composite figure.

*This is done and the old Fig. 1 in the revised text becomes Fig. 2 (see below).*

- Section 2.1 or elsewhere. I am wondering how the scaling between the tank simulation and the real Ionian Sea has been done for (turbulent) diffusion? I see no comments on that in the text, while I believe that it might be worth to discuss somewhere. Also, please add and discuss any other limitation or approximation of the tank experiments which are relevant for the presented experiments.

*We have introduced Appendix B (see text below) where we address the scaling for turbulent diffusion between the laboratory experiments and the Ionian Sea.*

*"Appendix B: Turbulent diffusion*

*The turbulent diffusion could be estimated both in the central deep and the slope domain. In both areas the turbulent diffusivity can be associated with a stratified shear layer. The energy of a typical eddy of size L in this shear layer is of the order $E_t \sim (\varepsilon L)^{2/3}$, with $\varepsilon$ being the dissipation rate*

*in $m^2s^{-3}$, whereas the energy associated with buoyancy and shear is $E_b \sim (NL)^2$ and $E_s \sim (\partial u/\partial z L)^2$, respectively. Here $N^2 = -g(\partial \rho/\partial z)/\rho$ is the Brunt-Väisälä frequency and $\partial u/\partial z$ is the vertical shear.*

*Balancing turbulent and forcing components yields a buoyancy length (i.e., the Ozmidov length scale) $L_o = (\varepsilon N^{-3})^{1/2}$ and a shear length $L_s = (\varepsilon <\partial u/\partial z>^{-3})^{1/2}$. This latter scale was defined initially by Corrsin (1958) as the smallest scale at which anisotropy effects resulting from a large-scale shear are carried out by the turbulent cascade.*

*For low Richardson numbers the effect of shear dominates the effect of buoyancy, therefore the relevant quantity to define the mixing scale is the shear, while for large Richardson numbers the relevant quantity is the Brunt-Väisälä frequency N. The smaller of these lengths limits the typical eddy size.*

*Odier et al. (2012) proposed a model based on the Prandtl mixing length model with characteristic mixing length $L_m$ to relate the turbulent eddy diffusivity $v_t$ to the velocity fluctuations and gradients in a stratified shear layer. They showed that $L_m$ was proportional to $L_s$ so that the turbulent diffusivity can be evaluated using:*

*$v_t \sim L_s^2 <\partial u/\partial z>$.*

*In the tank experiments, we estimate, $\varepsilon \approx O(10^{-2} \, m^2s^{-3})$ from the PIV measurements, $N=0.1 \, s^{-1}$, $<\partial u/\partial z> \approx O(1 \, s^{-1})$. Introducing these values in the above expression we obtain for $L_o \approx 3$ m and for $L_s \approx 0.1$ m; hence, the length scale will be determined by shear since $L_o > L_s$ so that our estimate gives $v_t \sim 10^{-4} \, m^2s^{-1}$ for the tank experiments. Using a velocity scale, the Nof speed $U=0.1 \, ms^{-1}$ and the vertical scale of the gravity current of $h=0.05$ m, the normalized turbulent diffusivity gives $v_t(Uh)^{-1} \sim 0.02$. We also expect that at higher Richardson numbers, as typical in the ocean, the length scale will be determined by buoyancy, since $L_o < L_s$, so that our oceanic estimate may be a bit high. Parametrizations in ocean models (Lane-Serff and Baines 2000) have used values in the range $0.032 < v_t < 0.70 \, m^2s^{-1}$ for typical overflow scenarios. Critical to extrapolating to oceanic conditions is a systematic exploration of the dependence of the mixing lengths on turbulence intensity and on the degree of stratification as measured by the Richardson number.*

*Lane-Serff and Baines (2000) also proposed a relation to evaluate the turbulent diffusivity based on scales that are easier to extrapolate for oceanic overflows, which reads:*

*$v_t = (4k^2Qf^2)(g's^2)^{-1}$,*

*where Q is the injected volume transport of the gravity current, k is taken to be $2.5 \times 10^{-3}$ (a typical value for oceanographic flows, e.g., Lane-Serff 1993, 1995; Bombosch and Jenkins 1995). This gives for the values of the Adriatic outflow into the Ionian Sea $Q \approx 10^5 \, m^3s^{-1}$, $f=10^{-4} \, s^{-1}$, $g'=0.003 \, ms^{-2}$ and $s=0.02$ giving $v_t \approx 0.02 \, m^2s^{-1}$. Rare in situ observations of upper ocean turbulent mixing, stratification and currents in the Adriatic Sea resulted in the estimates of the eddy diffusivity in the central part of the basin (Peters and Orlić, 2005). The values are however much smaller than those presented above due to weak shear and strong stratification combined with large Richardson numbers. Using $U=0.1 \, ms^{-1}$ as a typical velocity scale and a typical vertical length scale of the overflow of $h=200$ m, one obtains a normalized turbulent diffusivity of $v_t(Uh)^{-1} \approx 0.001$ for the real flow conditions in the Ionian Sea, which is smaller than the laboratory value (0.02). Note that if the expression of Lane-Serff and Baines (2000) for the rotating platform conditions is applied to oceanic environment, the value of the constant k needs to be adapted to*

*larger values, since the Reynolds number is smaller in the laboratory conditions than in the real ocean (see Lane-Serff 1993, 1995; Bombosch and Jenkins 1995).*

*1) Bombosch, A. and Jenkins A. : Modeling the formation and deposition of frazil ice beneath the Filchner–Ronne Ice Shelf. J. Geophys. Res., 100, 6983–6992, 1995.*
*2) Corrsin S. : Local isotropy in turbulent shear flow, National Advisory Committee for Aeronautics RM 58B11, 1958.*
*3) Lane-Serff, G. F. : On drag-limited gravity currents.,Deep-Sea Res. I, 40, 1699–1702, 1993.*
*4) Lane-Serff, G. F. : On meltwater under ice-shelves, J. Geophys. Res., 100, 6961–6965, 1995.*
*5) Lane-Serff, G. F. and P. G. Baines: Eddy formation by overflows in stratified water. J. Phys. Ocean., 30, 327–337, 2000.*
*6) Odier, P., Chen J. and R. E. Ecke:  Understanding and modeling turbulent fluxes and entrainment in a gravity current, Phys. D: Nonlin. Phen., 10.1016/j.physd.2011.07.010, 2012.*
*7) Peters, H. and M. Orlić: Turbulent mixing in the springtime central Adriatic Sea. Geofizika, 22, 1-19, 2005."*

- Figure 2 caption. Please add the area or the location for which density and MLD is presented (at the very centre, Cp3 ?)

*The location of the sensor used to measure density is presented in the new Fig. 1 (see the diamond symbol at the panel b) as follows:*

[Figure]

[Figure]

***Figure 1: (a) Map of the study area in the Ionian basin with a simplified circulation scheme, which changes accordingly to the BiOS regime. Grey horizontal lines indicate the geographical limits within which the mean vorticities above and below the 2200 m isobath were calculated. Rectangles A and B indicate the areas where density data (CMEMS reanalysis) were averaged. Concentric rings represent the simplified laboratory tank scheme. Acronyms: AW = Atlantic Water, LIW = Levantine Intermediate Water, AdDW = Adriatic Deep Water; (b) a view of the tank: the slope area is between the red and blue, deep flat-bottom area is inside the blue ring. Dense water injectors are placed at IS1 and IS2. A diamond near the centre shows a location of the Cp3 profiler. Concentric grey rings indicate intersections of the laser sheet levels with the slope. Grey dots indicate a regular x-y grid for***

*tank velocity field (subsampled every 5 nodes for clarity). The map in (a) was created from the bathymetry data ETOPO2v2, NOAA, World Data Service for Geophysics, Boulder, June 2006, doi: 10.7289/V5J1012Q) using the MATLAB software.*

[Figure]

**Figure 2: Scheme of the rotating tank (not in scale) and density configuration for the three experiments discussed in the paper, EXP 24, EXP 26 and EXP 27. Left hand side: cross section with a central deep area, a slope and injectors IS1 and IS2; blue/cyan patches refer to the lower/upper layers; numbers from 1 to 12 indicate inclined laser sheet levels. Right hand side: initial density in the lower/upper layers (blue/cyan lines); density and discharge rate of the injected water (red/black colour from IS1/IS2); the thickness of the red and black lines corresponds to discharge rates during various phases (for details see Tables 1 and 2). Only EXP 27 has both injectors active.**

- Line 377. Densities of 2010 and 2015 kg/m3 (?)

*Sorry for the mistake, corrected into 1010 and 1015.*

- Fig. 9. There are two (b) marks in the figure - please correct.

*In a new version of the manuscript the mistake was corrected*

- Lines 434-436. Why did you took 2200 m as the borderline between the slope and the flat central region? Just by looking in the topography 3000 m looks more appropriate for me (plus moving lower boundary line more to the south). Please justify your choice.

*We took the 2200 m isobath as the limit between the slope area and the open sea after several attempts with other choices and noticed that, essentially, all limiting isobaths show similar features. We finally end up with taking 2200 m since in that case the number of points for both open-sea and slope areas and the number of vorticity points are similar, and the statistical significance of the average vorticity is equally representative.*

- Line 566. If you have doi, you don't need to provide the direct link to the reference.

- Line 572. Missing "doi:"

- References. Please unify: doi or DOI or https://doi.org/... and add missing doi numbers for all references (for these which have them).

- References. Please unify: short or full journal names.

*The references were corrected accordingly in a new version of the manuscript.*

---

## Author Comment (AC4)

**Response to the comments of the Ref. 2**

The manuscript presents a reproduction of an important quasi-decadal mechanism that drives thermohaline oscillations in the Adriatic-Ionian region. I appreciate such a novel approach using tank experiments, which tries to provide the underlying physics that has (still) several possible explanations - this research is important to put proper weight on them (internal forcing vs. wind-stress curl) Therefore, I strongly recommend publication of the manuscript. Still, some issues should be cleared and corrected (also agreeing with comments of Anonymous Referee #1):

- Lines 94-106. It might be more appropriate to have this at the beginning of Section 2 (as an introduction, before Section 2.1), as justifying the applied methodology.

*As suggested by the referee, the text between lines 94 and 106 (see below) was moved after the second paragraph of Section 2.1. :*

*"We distinguish the slope and central deep (flat bottom) areas in the tank that are equivalent to the continental slope and deep zone of the northern Ionian basin, respectively (Fig. 1). We compare the potential vorticity evolution in each area as related to the dense water flow. The two areas are presumably controlled by different processes of the vorticity generation. In the central area (flat bottom) the upper layer squeezing, due to the downslope sinking of the dense water to the lower layer, generates the upper layer anticyclonic vorticity. In the slope area, the upper layer stretching due to the downslope water flow results in the generation of the cyclonic vorticity. The lower layer on the slope is subject to squeezing and anticyclonic vorticity generation as related to the formation of the dense water flow parallel to isobaths."*

- Fig. 1. It might be good to increase the font of the smallest labels, they can be hardly read in such a composite figure.

*This is done and the old Fig. 1 in the revised text becomes Fig. 2 (see below).*

- Section 2.1 or elsewhere. I am wondering how the scaling between the tank simulation and the real Ionian Sea has been done for (turbulent) diffusion? I see no comments on that in the text, while I believe that it might be worth to discuss somewhere. Also, please add and discuss any other limitation or approximation of the tank experiments which are relevant for the presented experiments.

*We have introduced Appendix A (see text below) where we address the scaling of the turbulent diffusion and comparison between the laboratory experiments and the Ionian Sea.*

**"Appendix A:**

[revised manuscript text omitted]

*1) Bombosch, A. and Jenkins A. : Modeling the formation and deposition of frazil ice beneath the Filchner–Ronne Ice Shelf. J. Geophys. Res., 100, 6983–6992, 1995.*
*2) Corrsin S. : Local isotropy in turbulent shear flow, National Advisory Committee for Aeronautics RM 58B11, 1958.*
*3) Lane-Serff, G. F. : On drag-limited gravity currents.,Deep-Sea Res. I, 40, 1699–1702, 1993.*
*4) Lane-Serff, G. F. : On meltwater under ice-shelves, J. Geophys. Res., 100, 6961–6965, 1995.*
*5) Lane-Serff, G. F. and P. G. Baines: Eddy formation by overflows in stratified water. J. Phys. Ocean., 30, 327–337, 2000.*
*6) Odier, P., Chen J. and R. E. Ecke: Understanding and modeling turbulent fluxes and entrainment in a gravity current, Phys. D: Nonlin. Phen., 10.1016/j.physd.2011.07.010, 2012.*
*7) Peters, H. and M. Orlić: Turbulent mixing in the springtime central Adriatic Sea. Geofizika, 22, 1-19, 2005."*

- Figure 2 caption. Please add the area or the location for which density and MLD is presented (at the very centre, Cp3 ?)

*The location of the sensor used to measure density is presented in the new Fig. 1 (see the diamond symbol at the panel b) and specified in the new Fig. 2 as follows:*

[Figure]

[Figure]

*Figure 1: (a) Map of the study area in the Ionian basin with a simplified circulation scheme, which changes according to the BiOS regime. Gray horizontal lines indicate the geographical limits within which the mean vorticities above and below the 2200 m isobath were calculated. Rectangles A and B indicate the areas where density data (CMEMS reanalysis) were averaged. Concentric rings represent the simplified laboratory tank scheme. Acronyms: AW = Atlantic Water, LIW = Levantine Intermediate Water, AdDW = Adriatic Deep Water; (b) Top view of the tank: the slope area is between the red and blue circles, the deep flat-bottom area is inside the blue ring. Dense water injectors are placed at IS1 and IS2. A diamond near the center shows the location of the Cp3 profiler. Concentric gray rings indicate intersections of the laser sheet levels with the slope. Gray dots indicate regular x-y grid nodes for the tank velocity field (subsampled every 5 nodes for clarity). The map in (a) was created from the bathymetry data ETOPO2v2, NOAA, World Data Service for Geophysics, Boulder, June 2006, doi: 10.7289/V5J1012Q) using the MATLAB software.*

[Figure]

*Figure 2: Scheme of the rotating tank (not to scale) and density configuration for the three experiments discussed in the paper, EXP 24, EXP 26, and EXP 27. Cross sectional view with central deep and slope areas, and injectors IS1 and IS2 (a, c, e); a vertical bar near the center of the tank indicates location of the vertical profiler Cp3; blue/cyan patches refer to the lower/upper*

*layers; numbers from 1 to 12 indicate inclined laser sheet levels. Initial density (blue/cyan bars) in the lower/upper layers along with density and discharge rates of the injected water (b, d, e); red/black bars correspond to IS1/IS2; the thickness of the bars corresponds to discharge rates during various phases (for details see Tables 1 and 2). Only in EXP 27 were both injectors active.*

- Line 377. Densities of 2010 and 2015 kg/m3 (?)

*Sorry for the mistake, corrected into 1010 and 1015.*

- Fig. 9. There are two (b) marks in the figure - please correct.

*In a new version of the manuscript the mistake was corrected (See new Fig. 10).*

- Lines 434-436. Why did you took 2200 m as the borderline between the slope and the flat central region? Just by looking in the topography 3000 m looks more appropriate for me (plus moving lower boundary line more to the south). Please justify your choice.

*We took the 2200 m isobath as the limit between the slope area and the open sea after several attempts with other choices and noticed that, essentially, different isobaths show similar features. We finally end up with taking 2200 m since in that case the number of points for both open-sea and slope areas and the number of vorticity points are similar, and the statistical significance of the average vorticity is equally representative.*

- Line 566. If you have doi, you don't need to provide the direct link to the reference.

- Line 572. Missing "doi:"

- References. Please unify: doi or DOI or https://doi.org/... and add missing doi numbers for all references (for these which have them).

- References. Please unify: short or full journal names.

*The references were corrected accordingly in a new version of the manuscript.*

---

## Author Comment (AC5)

**Response to the comments of the Ref. 3**

C. 1: The manuscript encompassing a comprehensive introduction of the experimental apparatus and methodology followed by the authors and everything is very well conducted, except the relative role (scale ratio) of the central part of the tank respect to the sloping part. In fact, looking the figure 7 in the manuscript seems that the dynamics driven by the slope domain dominates on those generate in the flat domain, making difficult distinguishing the difference between the two dynamics. This isn't irrelevant to make more realistic the comparison with the northern Ionian Sea circulation in section 4.

*Reply: We thank the Referee for the comment on the relative importance of the dynamics in the slope domain with respect to the dynamics of the flat domain. Regarding this, first we bring attention to the fact that Fig. 7 (Fig. 8 in the revised version of the manuscript) shows only the surface layer flow evolution in the tank. The dense water injection occurs over the slope near the interface between the upper and lower layers and begins influencing the stretching and squeezing of the water column in different manner. We recall Fig. 5, in particular Figs. 5b and 5d of the revised version (old Fig. 4) and Fig. 11. These figures show how the averaged vorticity in the slope and flat domains evolves in time, both in the tank and the Ionian Sea. Concerning the relative importance of the slope with respect to the central flat-bottom area, in the Chapter 4 we compare residence times in the Ionian Sea and in the rotating tank. The ratio between the two residence times is of the same order of magnitude as the ratio of the vorticity rate of change in the rotating tank and in the Ionian Sea. This confirms the dynamic similarities of the dense water flow in the two basins making realistic the comparison between the laboratory experiment and the Ionian Sea. Furthermore, this confirms that the relative importance of processes at the slope and in the central part of the basin is similar in the Ionian and the rotating tank. Indeed, our main goal is to show that laboratory conditions are similar to the ocean processes and not to distinguish between the slope and flat-bottom phenomena which are obviously interdependent.*

*To clarify the question of dynamic similarity, we added the following text at the end of chapter 4:*

*"The dynamic similarity between the Ionian Sea and the laboratory experiments was discussed in Rubino et al. (2020) by comparing the Burger number in the laboratory and oceanic conditions. Also, as evidenced in the same paper, another important similarity that should exist is between the in situ and laboratory ratios of the topographic slope and the initial geostrophic slope, which means that the non-dimensional number $g's(fV)^{-1}$, must be preserved in the laboratory."*

C2 Moreover is very confusing the theoretical and modelling equations that are used to analyze the experimental results: the equation 1 is not the same used by the cited paper of Lee-Lueng, F. and Davidson, R. A. (A note on the barotropic response of sea level to time-dependent wind forcing. J. Geophys. Res., 100, C2, 24955-24963, 1995) that use a classical linear barotropic vorticity equation, may be the authors have to use a different reference.

*Reply: As our response to this comment, we simply removed the reference saying that we will be treating the well-known linear barotropic vorticity equation for an f-plane approximation with the sloped bottom in radial coordinates for the surface layer without wind-stress forcing. We also defined the radial coordinate being perpendicular to isobaths and negative downslope.*

C3 However, the most relevant matter is related to the stratification that, at the end, is the core problem of the manuscript. It is well know that a good representation of the ocean dynamics is a three-layer system and this is this is confirmed even in this case as is well evident in the figure 2c (experiment 27), specifically around the 75th day in which we see the respond of the pressure to the injection of the density anomaly and subsequence stratification in three layer (or a continuously stratification see references), is very interesting the impact of the redistribution of density and pressure within the water column in the figure 3 (and also figure 5) experiment 27 at the same day (around the 75th). These figures are the most interesting of the manuscript and at the same time are those that demonstrate the weakness of the theory presented in this manuscript: actually, can't demonstrate the opposite vorticity at the 75th day and the corresponding kinetic energy anomaly in the lower layers. However, despite this experimental evidence and the same recognition as the authors themselves that the dynamics follow at least a two-layers system, even so at the end the equation that the authors used is written in a one-layer formulation. This is not irrelevant for physical point of view. Is matter of fact that dealing with one-, two- or three-layer formulation of the QG equation, produce a different vorticity relation between the several interfaces along the water column (see Sokolovskiy paper and all reference herein). This is true either in the flat or in the slope domain and finally on the comparison with the realistic example of the Ionian Sea. In conclusion, the circular rotating tank experiment shows in an impressing way (this could be more impressing with a different scale ratio between the slope/flat domain), the adjustment of the vorticity along the continuously stratified water column (and its dependence from the layers-thickness distribution) when it is subjected to a density anomalies: first along the slope and afterwards during the spreading of the anomaly density flow along the flat bottom; finally is very arduous to do some comparison, in the present version of the manuscript, between the tank experiment and what was observed in the northern Ionian sea in 2012.

Reply: *The Reviewer is right when he argues that the evolution of a 2-layer and 3-layer system differs. However, in the present paper we do not attempt to determine the evolution of the eddying dynamics of the system. Our focus is on the dynamics of two homogeneous layers and the relationship between their thickness and the relative vorticity which is constrained for every layer by the conservation of potential vorticity (PV) in each of them, independently of the dynamics above and below it. So, the conservation of PV is ensured in a single layer, and it does not depend on how many layers stay above or below. In addition, we reduce the effect of the horizontal advection in our analysis by considering horizontal averages over larger areas (central and slope). Please, note also that the flow is not advected from one layer to the other.*

*We thank the Reviewer for highlighting the event of the 75$^{th}$ day in experiment 27 which is really a special event. We examined more carefully the flow pattern evolution during that event and added Appendix B with details.*

*"Appendix B:*
*The theoretical curve in Fig. 4 obtained from Eq. 3 on the lower layer thickness h fits rather well the experimental data from the single-point density measurements at Cp3 site. Few departures from the theoretical curve are however evident, like e.g. the anomaly seen in the MLD (Fig. 3, reported also in Fig. B1a), in the rate of change of the lower layer thickness (Fig. 4), and in the vorticity evolution (Fig. 6) for EXP 27, around rotation day 75$^{th}$. Concurrently, at the Cp3 site, the*

*local maximums are present both in the radial and tangential velocities in the deep layer (Figs. B1 b and c). These features are linked to the passage of a mesoscale anticyclonic eddy. Indeed, the formation and passage of the eddy in the vicinity of Cp3 is clearly evident from the horizontal distributions of the velocity vectors in the lower layer beneath the pycnocline, represented by the level 11 (Fig. B2). Overall effects of the eddy passage are: an anomaly in the vertical density distribution and the transient thickening of the pycnocline layer, an anomaly in the relationship between the vorticity rate of change and the rate of change of the bottom layer thickness (Figs. 3 and 4). This event is however of limited spatial and temporal extent. In fact, it should be kept in mind that such an anomaly has been detected because the mesoscale eddy was passing through the Cp3 site. Moreover, less prominent features are observed at other rotation days, probably generated by similar mesoscale features passing close but not over the Cp3 location. It is plausible that some similar eddies have not been detected at all, because travelling out of range of Cp3 site, during EXP 27, but also during other experiments."*

[Figure]

*Figure B1: Hovmöller diagram of density (a), radial (b) and tangential (c) velocity components for EXP 27. The black isoline interval in panel (a) is 2 kg m⁻³, starting from 1016 kg m⁻³ at the bottom; thick cyan and blue lines denote MLD and the base of the pycnocline, respectively. Bold isolines in panels (b) and (c) indicate 0 cm s⁻¹, and the isoline interval is 0.5 cm s⁻¹. Black solid vertical lines indicate rotation days 45 and 90, i.e., the start and end of the high-density water injection, respectively (for reference see Fig. 2 and Tables 1 and 2).*

[Figure]

*Figure B2: Evolution of the flow field in the lower layer of the rotating tank (represented by level 11) during experiment 27. The rotation days are indicated inside each panel. The diamond shows the location of the conductivity probe Cp3, and the two bars show the location of the two dense water sources. The outer and inner bold circles delimit the tank edge and the central deep area, respectively. Grey circle delimits the extension of the level 11.*

Minor revision:

· Line 326 "level 1" is referred to inclined laser sheet levels 1 of the Figure 1?

*Reply: Yes, it is.*

· Line 451 at which model is referred? Please give more details;

*Reply: We clarified this point in section 2 Data and methods:*

*"We compare the current fields in the rotating tank and in the real ocean for a particular condition when a circulation inversion event was observed in the northern Ionian Sea. Regarding the real ocean, for the surface we use the altimetry data, while for the deep layer conditions we use outputs from the hydrodynamic model of the Mediterranean Forecasting System. The latter concerns the physical reanalysis component, originating from the Copernicus Marine Service MEDSEA_REANALYSIS_PHYS_006_004 dataset (CMEMS Reanalysis) supplied by the Nucleus for European Modelling of the Ocean (NEMO) (Simoncelli et al., 2019). The model has a horizontal grid resolution equal to 1/16˚ (ca. 6-7 km) and 72 unevenly spaced vertical levels. We use the following variables: 3D monthly mean and daily mean temperature, salinity, and horizontal current components (eastward and northward) covering the entire Mediterranean Sea (https://doi.org/10.25423/MEDSEA_REANALYSIS_PHYS_006_004 https://resources.marine.copernicus.eu/?option=com_csw&view=details&product_id=MEDSEA_MULTIYEAR_PHY_006_004)."*

*From then on, we specify everywhere in the text that we deal with the CMEMS Reanalysis fields, including the Line 451:*

*"By comparing the outputs of the CMEMS Reanalysis (which assimilates the in-situ data) and the vertical profiles of those ARGO floats, that were active in the northern Ionian during 2012, we observe that the Reanalysis absolute density values are typically larger than float densities (not shown). However, temporal variations of both data sets are consistent. Thus, we reconstruct the evolution of the density fields on the continental slope and in the deep area of the northern Ionian Sea using only the data from the CMEMS Reanalysis and compare it with the vorticity variations at the surface and in the deep layer (i.e., 1000 m depth, Fig. 12)."*

· figure 1 the word "cp3" is not clear;

*Reply: The meaning and location of the Cp3 is now introduced in the new Figure 1 (see below). In addition, the caption of the old Fig. 1 (now Fig. 2) also indicates the meaning of the Cp3.*

[Figure]

*Figure 1: (a) Map of the study area in the Ionian basin with a simplified circulation scheme, which changes according to the BiOS regime. Gray horizontal lines indicate the geographical limits within which the mean vorticities above and below the 2200 m isobath were calculated. Rectangles A and B indicate the areas where density data (CMEMS reanalysis) were averaged. Concentric rings represent the simplified laboratory tank scheme. Acronyms: AW = Atlantic Water, LIW = Levantine Intermediate Water, AdDW = Adriatic Deep Water. (b) Top view of the tank: the slope area is between the red and blue circles, the deep flat-bottom area is inside the blue ring. Dense water injectors are placed at IS1 and IS2. A diamond near the center shows the location of the Cp3 profiler. Concentric gray rings indicate intersections of the laser sheet levels with the slope. Gray dots indicate regular x-y grid nodes for the tank velocity field (subsampled every 5 nodes for clarity). The map in (a) was created from the bathymetry data ETOPO2v2, NOAA, World Data Service for Geophysics, Boulder, June 2006, doi: 10.7289/V5J1012Q) using the MATLAB software.*

[Figure]

*Figure 2: Scheme of the rotating tank (not to scale) and density configuration for the three experiments discussed in the paper, EXP 24, EXP 26, and EXP 27. Cross sectional view with central deep and slope areas, and injectors IS1 and IS2 (a, c, e); a vertical bar near the center of the tank indicates location of the vertical profiler Cp3; blue/cyan patches refer to the lower/upper layers; numbers from 1 to 12 indicate inclined laser sheet levels. Initial density (blue/cyan bars) in the lower/upper layers along with density and discharge rates of the injected water (b, d, e); red/black bars correspond to IS1/IS2; the thickness of the bars corresponds to discharge rates during various phases (for details see Tables 1 and 2). Only in EXP 27 were both injectors active.*

figure 2 in the color tab 0-15 means that the range of density is between 1000-1015?

*Reply: The Reviewer is right, and figures have been re-done using density instead of density anomalies (see new Fig. 3)*